# An empirical data analysis of "price runs" in daily financial indices: Dynamically assessing market geometric distributional behavior

Héctor Raúl Olivares-Sánchez[1]⦿, Carlos Manuel Rodríguez-Martínez⦿[2]⦿, Héctor Francisco Coronel-Brizio[2]⦿, Enrico Scalas[3]⦿, Thomas Henry Seligman[4,5]⦿, Alejandro Raúl Hernández-Montoya⦿[2,6]⦿*

**1** Department of Astrophysics, Radboud University, Nijmegen, The Netherlands, **2** Instituto de Investigaciones en Inteligencia Artificial, Universidad Veracruzana, Xalapa Veracruz, México, **3** Department of Mathematics, School of Mathematical and Physical Sciences, University of Sussex, Brighton, United Kingdom, **4** Centro Internacional de Ciencias AC, Campus UAEM-UNAM, Cuernavaca Morelos, México, **5** Instituto de Ciencias Físicas, Universidad Nacional Autónoma de México, Cuernavaca Morelos, México, **6** Facultad de Física, Universidad Veracruzana, Xalapa Veracruz, México

⦿ These authors contributed equally to this work.
* alhernandez@uv.mx

**Data Availability Statement:** Data was available in www.yahoo https://es-us.finanzas.yahoo.com.

## Abstract

In financial time series there are time periods in which market indices values or assets prices increase or decrease monotonically. We call those events "price runs", "elementary uninterrupted trends" or just "uninterrupted trends". In this paper we study the distribution of the duration of uninterrupted trends for the daily indices DJIA, NASDAQ, IPC and Nikkei 225 during the period of time from 10/30/1978 to 08/07/2020 and we compare the simple geometric statistical model with $p = \frac{1}{2}$ consistent with the EMH to the empirical data. By a fitting procedure, it is found that the geometric distribution with parameter $p = \frac{1}{2}$ provides a good model for uninterrupted trends of short and medium duration for the more mature markets; however, longest duration events still need to be statistically characterized. Estimated values of the parameter $p$ were also obtained and confirmed by calculating the mean value of $p$ fluctuations from empirical data. Additionally, the observed trend duration distributions for the different studied markets are compared over time by means of the Anderson-Darling (AD) test, to the expected geometric distribution with parameter $p = \frac{1}{2}$ and to a geometric distribution with a free parameter $p$, making possible to assess and compare different market geometric behavior for different dates as well as to measure the fraction of time runs duration from studied markets are consistent with the geometric distribution with $p = \frac{1}{2}$ and in parametric free way.

## Introduction

Financial-market analysis studies the movements of price assets and financial indices. Extracting a profit from these movements is an important activity in the financial industry; a large variety of methods that intend to predict market behavior have been developed over the years,

Mathematica software and all data is also available in: https://github.com/CarlosManuelRodr/TrendDurationAnalysis and https://github.com/CarlosManuelRodr/TrendDurationAnalysis/tree/main/Research/OriginalDataset Resoectively.

**Funding:** ARHM and CMRM received support from grants 425854 and 5150 from the Consejo Nacional de Ciencia y Tecnología. Conacyt. https://conacyt.mx/ THS received support from grant number 425854 from the Consejo Nacional de Ciencia y Tecnología. Conacyt. https://conacyt.mx/ ES is partially supported by the Dr Perry James (Jim) Browne Research Centre at the Department of Mathematics, University of Sussex. http://www.sussex.ac.uk/broadcast/read/55282 The funders had no role in study design, data collection and analysis, decision to publish, or preparation of the manuscript.

**Competing interests:** The authors have declared that no competing interests exist.

ranging from complex mathematical models to even pseudo-scientific techniques [1]. An important approach is the statistical analysis of large sets of data, now partially available to small investors, as well, due to the increasing availability of computer power and high quality data sets. This analysis has benefited from the contributions not only from economists, but also from many physicists and mathematicians who have applied methods and ideas of probability theory and statistical physics to finance. As an academic result of these efforts, a set of universal, nontrivial statistical properties of financial historical data, persistent over time, has been observed and called "stylized facts" [2, 3].

When looking at price values of an asset on a financial time series chart, it is common to observe "price trends" in which most of the values are larger (or smaller) than the previous ones, these trends can be seen as composed by uninterrupted elementary trends, with periods in which the value increases or decreases monotonically. Trends are a popular subject within the so-called *technical analysis*. According to the followers of technical analysis, the *chartists*, patterns in the trend direction of financial data are believed to be indicators of changes in market direction and indicative of future behavior of prices. The effectiveness of this approach to financial markets is disputed and put at a stake by what is known as the Efficient Market Hypothesis (EMH), which indicates that current prices reflect available information. Elementary uninterrupted trends are the main subject of the present work, where we study empirically a basic random process consistent with the EMH allowing us to quantify trend directions in financial time series. From now on, we call these elementary uninterrupted trends only "trends", or more specifically, "uptrends" or "downtrends", depending of their direction. Empirical studies of financial and economic data are becoming increasingly relevant for the following reasons:

1) Currently dozens of stylized facts have been observed and more are still being discovered. 2) The study and prediction of stylized facts by means of methodologies of multi-agents market models is an important area of research in Finance and Econophysics. 3) Stylized facts are an import tool to validate proposed numerical and multi-agent market models; and 4) At present, we still lack a general, microscopic theory or model to explain the origin of stylized facts, we think simulation methodologies using agents could be useful in the construction of such a general theory. Some interesting references on these issues are the following: [3–10].

Before going further, it is necessary to present some preliminary and basic definitions. In subsections Definitions and The Efficient Market Hypothesis these definitions and other useful information will be presented. In section An 'Efficient Market' toy model for the distribution of run durations, a model for the distribution of trends duration will be developed consistently with the EMH. Section Data sample and methodology will explain how the data were analyzed and section Data analysis will provide an interpretation of the analysis.

## Definitions

Given a financial time series of asset prices or index values, $S(1), S(2), \ldots, S(n)$, let $X(t) = \log S(t)$ be the logarithm of its terms, where $t = 1, 2, \ldots, n$. A common quantity used to study price variations in financial time series is the *log-return* defined at time $t$ as

$$r(t, \Delta t) \approx X(t + \Delta t) - X(t) \tag{1}$$

for a given time sampling scale $\Delta t$. If the price variation is small, the log-return is a good approximation of the return

$$R(t, \Delta t) = \frac{S(t + \Delta t) - S(t)}{S(t)}. \tag{2}$$

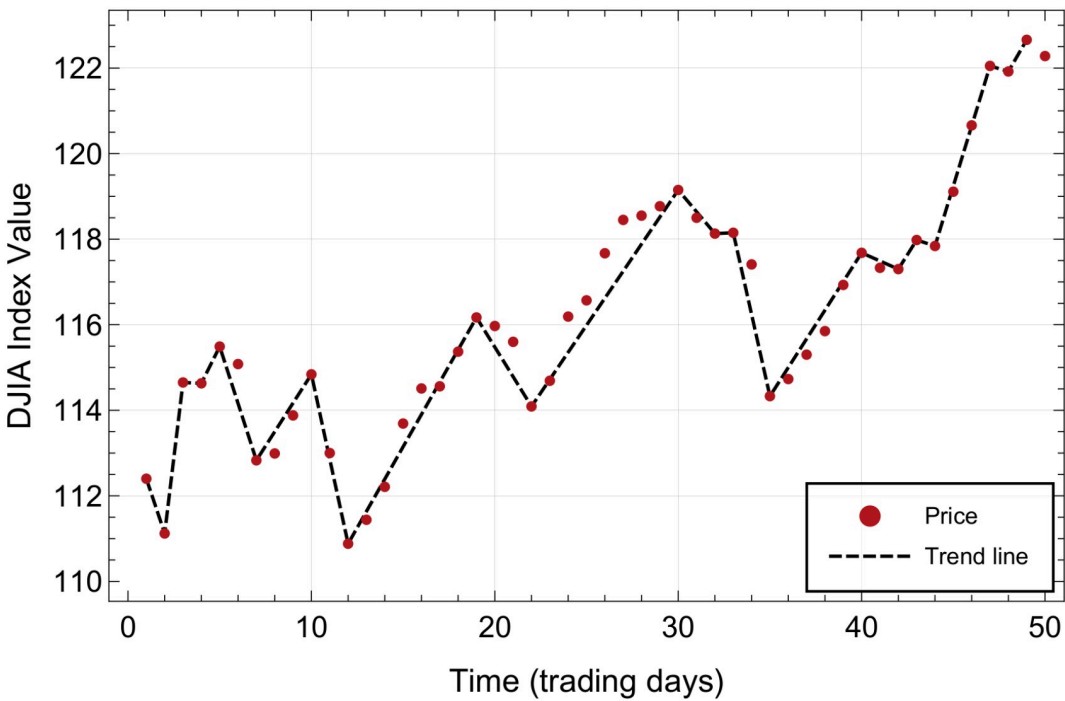

**Fig 1. Elementary trends on the time series for the prices of the DJIA during the period from Oct/30/1978 to Jan/09/1979.** The dashed line segments join the starting and ending points of each elementary trend.

In this paper, we consider $\Delta t$ equal to 1 day and we use the values of the indices corresponding to the close value in the investigated markets. More details on the data set will be given in section Data sample and methodology.

An *elementary trend* of duration $k$ is defined as a subseries of $k + 1$ values within the given time series $S(t)$ in which every value is greater (for an uptrend) or smaller or equal (for a downtrend) than the preceding one, an example of which is shown in Fig 1 for the prices of the DJIA, in a time period between October 1978 and January 1979. The duration of an elementary downward/upward trend in daily data is the number of days before the price changes direction, as the price varies, i.e. if the price does not change sign from one day to the other, the corresponding trend continues. In this figure and focusing our attention on red points, we see first an uninterrupted downtrend one day long, followed by a three days long uptrend, then a downtrend with a duration of two days, a three days long uptrend, etc. By construction, uptrends and downtrends appear alternately in the original time series $S(t)$.

Here, we present a detailed statistical study of these short elementary trends using market closing price values from four different indices over a time sampling scale of $\Delta t = 1$ day for the period between October 30, 1978 and August 07, 2020.

## The Efficient Market Hypothesis

The Efficient Market Hypothesis (EMH), first stated by Eugene F. Fama in 1970 [11], claims that the market quickly finds the rational price for a traded asset [12], as the current value incorporates all possible information about the price in the future. The most important consequence of this hypothesis was shown by P. Samuelson [13] and it is the fact that the best

forecast for the future price of an asset is its present price.

$$\mathbb{E}(S(t + \Delta t)|\mathcal{F}_t) = S(t),\qquad(3)$$

where $\mathbb{E}(\cdot|\mathcal{F}_t)$ is the conditional expectation with respect to the filtration $\mathcal{F}_t$, namely with respect to the known history up to time $t$. Indeed, it is easy to derive the above form of EMH starting from a simple statistical no-arbitrage argument. Suppose we have two assets, a risky one, with price $S(t)$ and a risk-free one giving a constant interest rate $r_F$. To avoid arbitrage, one has to require that the expected return of the risky asset is equal to the risk-free interest rate, that is

$$\mathbb{E}(R(t, \Delta t)|\mathcal{F}_t) = r_F;\qquad(4)$$

where $R(t, \Delta t)$ was defined in Eq (2), assuming for simplicity that no dividends are paid in the time interval $\Delta t$. The latter equation immediately yields, for non vanishing $S(t)$,

$$\mathbb{E}(S(t + \Delta t)|\mathcal{F}_t) = (1 + r_F)S(t),\qquad(5)$$

which reduces to Eq (3) for $r_F = 0$. Eqs (3) and (5), jointly with the integrability of the process $S(t)$, are known as martingale and sub-martingale conditions (remember that, under normal conditions $r_F \geq 0$, even if interest rates can be negative), respectively. From a technical point of view, one has to further assume integrability of the price process ($\mathbb{E}[|S(t)|] < \infty$), together with Eqs (3) or (5), and Eq (5) together with integrability means that the discounted price is a martingale when $r_F > 0$. Please notice that Eqs (3) and (5) are not uniquely specifying a random process for $S(t)$, but one can prove that, if they hold, then returns must be uncorrelated. In financial data, square returns or absolute returns turn out to be correlated (with long-range correlations), but this stylized fact does not falsify the EMH even if it is the main reason for the popularity of ARCH/GARCH models in financial econometrics [14, 15].

The EMH invalidates the pretence of technical analysis to predict future prices or trends; in fact, in Samuelson's words, "there is no way of making an expected profit by extrapolating past changes in the futures price, by chart or any esoteric devices of magic or mathematics" [13].

## An 'Efficient Market' toy model for the distribution of run durations

Among all the possible statistical models that can describe price fluctuations, the geometric random walk is the simplest one. A geometric random walk is just a product of independent and identically distributed positive random variables. If the expected value of these variables is 1, then the geometric random walk is a martingale; otherwise, if the expected value is larger than 1, the geometric random walk is a submartingale. However, the geometric random walk hypothesis is neither necessary nor sufficient for an efficient market, as shown by many authors, among whom Leroy [16], Lucas [17] and Lo and Mackinlay [1]. Again, to see this point, it is enough to consider that Eq (5) allows for any (sub)-martingale model.

To study our trends, at each step of a time series of price or index values, there are three possible outcomes: increase, constant and decrease, but the second one does not change prices direction. Then we consider the two possible outcomes: either the time series increases or it does not increase. In an efficient market, the expected future price only depends on information about the current price, not on its previous history. Therefore, it should be impossible to predict the expected direction of a future price change given the history of the price process. In formula, from Eq (3) (after discounting for the risk-free rate), we have

$$\mathbb{E}(S(t + \Delta t) - S(t)|\mathcal{F}_t) = 0;\qquad(6)$$

therefore, if we consider the sign of the price change $Y(t, \Delta t) = \text{sign}(S(t + \Delta t) - S(t))$, which coincides with the sign of returns, we accordingly have

$$\mathbb{E}(Y(t, \Delta t)) = 0. \tag{7}$$

If the price follows a geometric random walk, then the series of price-change signs can be modeled as a Bernoulli process. This process could be biased to take the presence of a risk-free interest rate into account. To be more specific, let us consider a log-normal geometric random walk and let us use the assumption $\Delta t = 1$. Let $S_0$ be the initial price. The price at time $t$ will be given by

$$S(t) = S_0 \prod_{i=1}^{t} Q_i \tag{8}$$

where $Q_i$ are independent and identically distributed random variables following a log-normal distribution with parameters $\mu$ and $\sigma$. These two parameters come from the corresponding normal distribution for log-returns. As a direct consequence of the EMH in the form Eq (5), we have

$$\mathbb{E}(Q) = 1 + r_F, \tag{9}$$

whilst for a log-normal distributed random variable the expected value is

$$\mathbb{E}(Q) = e^{\mu} e^{\sigma^2/2}; \tag{10}$$

by combining these two equations the following dependence between the parameters is found:

$$\sigma = \sqrt{2(\log(1 + r_F) - \mu)} \tag{11}$$

which allows us to compare the parameters estimated from the distributions of the index price returns, as the values of the two parameters $\mu$ and $\sigma$ come from the corresponding normal distribution for log-returns of the price or index data. Further, from the cumulative distribution function of a log-normal random variable

$$F_Q(u) = \mathbb{P}(Q \leq u) = \frac{1}{2} + \frac{1}{2} \text{erf}\left(\frac{\log(u) - \mu}{\sqrt{2\sigma^2}}\right), \tag{12}$$

we find that the probability of a negative sign of the return is given by

$$q = F_Q(1) = \mathbb{P}(Q \leq 1) = \frac{1}{2} + \frac{1}{2} \text{erf}\left(\frac{\sigma}{2\sqrt{2}} - \frac{\log(1 + r_F)}{\sqrt{2}\sigma}\right). \tag{13}$$

For typical markets, from 1978 to 2020, the value of the daily risk-free rate of returns oscillated in the range $0 < r_F \leq 2.5 \times 10^{-4}$. Eq (11) can be tested using the values of $r_F$ and the estimates of $\mu$ and $\sigma$) as can be seen in Fig 2. Using these values for $r_F$, the probability of negative returns is found to be $q = 0.5 \pm 0.02$, thus the Bernoulli process seems a reasonable first approximation for the probability of a change in sign for the return data.

Under this framework, it becomes natural to use the biased Bernoulli process as the null hypothesis for the time series of sign changes of the log-returns [18]. It is known that the distribution of the number $x$ of failures needed to get one success for a Bernoulli process with success probability $p = 1 - q$ is the geometric distribution $\mathcal{G}(p)$. The number of failures is then

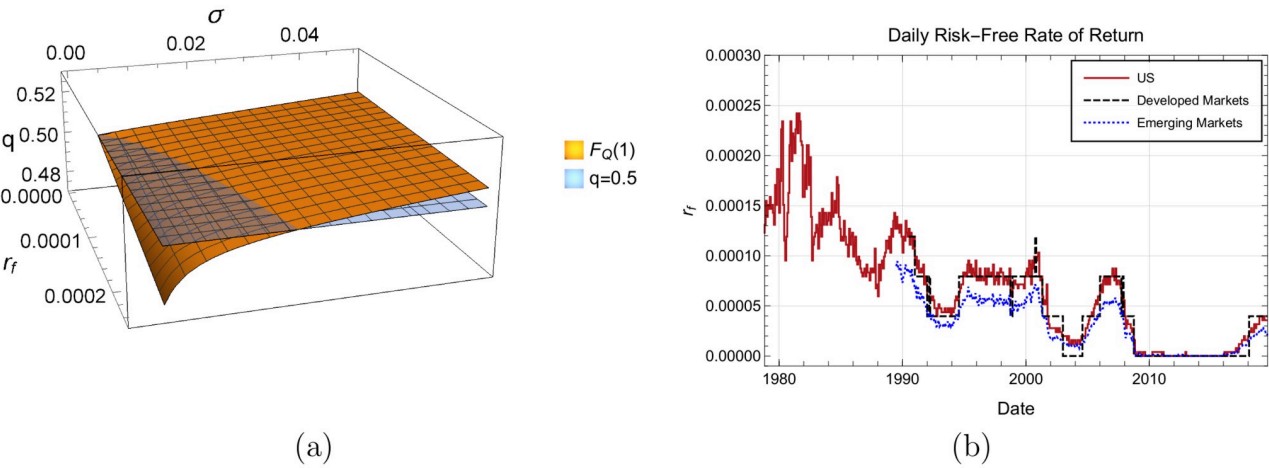

**Fig 2. For a typical time series of $Q_i$ the parameter $\mu$ is close to zero oscillating on a small interval ranging from −0.004 to 0.05.** The risk-free rate of return $r_F$ oscillates between the interval [0, 0.061]. For these values the probability of a negative return is $q = 0.5\pm0.02$. Subfig 2(a) is the probability of negative return given the risk-free interest rate $r_f$ and mean $\sigma$. The $\sigma$ parameter can be estimated from the return time series $Q_i$, and $r_f$ is estimated from a reference asset that depends on the market being studied. Intersection with $q = 0.5$ is also shown. Subfig 2(b) is the daily risk-free rate of returns $r_F$ for the US market, developed markets and emerging markets. For the period ranging from 1978 to 2020 the daily risk-free rate oscillates in the interval [0, 0.061]. Data was downloaded from http://mba.tuck.dartmouth.edu/pages/faculty/ken.french/data_library.html.

given by

$$P(x) = \mathbb{P}(N = x) = p(1 - p)^x = pq^x. \tag{14}$$

The duration of an elementary downward trend in daily data is the number of days before the price increases, so the distribution of such trend duration should follow a geometric distribution. An identical argument applies to the duration of an upward trend. Note that such sequences of identical outcomes are also known as *runs* or *clumps* in the mathematical literature. Some historical references on this subject, are [19–21]. Where chapter X of the first reference was during many years the classical textbook reference to Theory of Runs; second reference shows an interesting statistical test based on runs properties to demonstrate that two sets of independent observations corresponding to two independent random variables have the same distribution and finally, the third reference presents an intensive treatment of the theory of runs still of current interest.

In the next section we describe the data used for testing the model just presented, as well as a discussion of the goodness of fit test applied to compare the observed and expected distributions, namely the geometric distribution, of the duration of the trends of upwards/downwards price, which coincide with the sign of the log-returns.

## Data sample and methodology

In this work, daily close data values of four financial indices were analyzed, namely Dow Jones Industrial Average (DJIA), NASDAQ Composite, the Mexican Índice de Precios y Cotizaciones (IPC) and Nikkei 225, during the period between October 30 1978—August 07 2020. All data sample for the mentioned time span is available as suplementary material, see S1 Dataset at the end of this paper. Number of analyzed records and found uninterrupted uptrends and downtrends, as defined in subsection Definitions are displayed in Table 1.

**Table 1. Numbers of total observed records and respective uninterrupted trends for all data samples of financial indices studied.** The data have been filtered, e.g. by removing null records.

| Market | Records | Trends | Uptrends | Downtrends |
|---|---|---|---|---|
| DJIA | 10571 | 5362 | 2681 | 2681 |
| Nasdaq | 10534 | 4779 | 2389 | 2390 |
| IPC | 10432 | 4410 | 2205 | 2205 |
| Nikkei | 10300 | 5167 | 2583 | 2584 |

Remember that by construction, for each data sample, the number of uninterrupted uptrends and downtrends are the same if the analyzed financial time series has an even number of total trends and they differ in one unity if the total number of trends is odd respectively.

The composition of trends for each data sample is described in Tables 2–5. Additional and brief comments on the different duration of constructed uninterrupted trend data samples may be found in section Conclusions.

Finally, in Table 6, we show the descriptive statistics of data presented in the current section. Values of first four central moments are displayed. It can be seen that the mean value of the observable trends duration for all studied markets is close to two, it is bigger for less mature markets and that uptrends mean duration is slightly bigger that downtrend mean duration for all markets.

## The Anderson-Darling goodness of fit test

In order to compare the observed and expected distributions of trend durations, the Anderson-Darling (AD) test described in references [22, 23] was used. The AD test belongs to a family of goodness of fit tests called the Cramér-von Mises tests, which includes the Anderson-Darling test, Watson's test and the Cramér-von Mises test itself. The family was originally developed to test continuous distributions, but a generalization for discrete distributions appeared for the first time in an article by Choulakian *et.al.* [23]. The Anderson-Darling test was found to be the most suitable for this purpose because it places more weight on the tails of a distribution than other goodness of fit tests.

**Table 2. Composition of uninterrupted trends observed in the DJIA data sample.**

| Duration (days) | 2681 Uptrends | 2681 Downtrends | 5362 Total |
|---|---|---|---|
| 1 | 1262 | 1429 | 2691 |
| 2 | 674 | 673 | 1347 |
| 3 | 354 | 316 | 670 |
| 4 | 210 | 147 | 357 |
| 5 | 90 | 70 | 160 |
| 6 | 45 | 28 | 73 |
| 7 | 23 | 10 | 33 |
| 8 | 11 | 7 | 18 |
| 9 | 5 | 0 | 5 |
| 10 | 3 | 0 | 3 |
| 11 | 2 | 0 | 2 |
| 12 | 1 | 1 | 2 |
| 13 | 1 | 0 | 1 |

**Table 3. Composition of uninterrupted trends observed in the Nasdaq data sample.**

| Duration (days) | 2389 Uptrends | 2390 Downtrends | 4779 Total |
|---|---|---|---|
| 1 | 977 | 1240 | 2217 |
| 2 | 532 | 586 | 1118 |
| 3 | 384 | 300 | 684 |
| 4 | 209 | 128 | 337 |
| 5 | 120 | 71 | 191 |
| 6 | 58 | 36 | 94 |
| 7 | 45 | 13 | 58 |
| 8 | 29 | 10 | 39 |
| 9 | 10 | 4 | 14 |
| 10 | 9 | 1 | 10 |
| 11 | 7 | 0 | 7 |
| 12 | 5 | 0 | 5 |
| 13 | 1 | 0 | 1 |
| 14 | 1 | 0 | 1 |
| 15 | 0 | 0 | 0 |
| 16 | 0 | 1 | 1 |
| 17 | 0 | 0 | 0 |
| 18 | 1 | 0 | 1 |
| 19 | 1 | 0 | 1 |

**Table 4. Composition of uninterrupted trends found in the IPC data sample.**

| Duration (days) | 2205 Uptrends | 2205 Downtrends | 4410 Total |
|---|---|---|---|
| 1 | 865 | 981 | 1846 |
| 2 | 537 | 559 | 1096 |
| 3 | 320 | 323 | 643 |
| 4 | 192 | 151 | 343 |
| 5 | 111 | 97 | 208 |
| 6 | 74 | 42 | 116 |
| 7 | 45 | 25 | 70 |
| 8 | 22 | 8 | 30 |
| 9 | 16 | 8 | 24 |
| 10 | 10 | 4 | 14 |
| 11 | 5 | 3 | 8 |
| 12 | 3 | 1 | 4 |
| 13 | 0 | 1 | 1 |
| 14 | 1 | 0 | 1 |
| 15 | 3 | 0 | 3 |
| 16 | 0 | 0 | 0 |
| 17 | 0 | 0 | 0 |
| 18 | 0 | 0 | 0 |
| 19 | 0 | 1 | 1 |
| 20 | 1 | 0 | 1 |
| 21 | 0 | 0 | 0 |
| 22 | 0 | 0 | 0 |
| 23 | 0 | 1 | 1 |

**Table 5. Composition of uninterrupted trends observed in Nikkei index.**

| Duration (days) | 2583 Uptrends | 2584 Downtrends | 5167 Total |
|---|---|---|---|
| 1 | 1253 | 1337 | 2590 |
| 2 | 652 | 644 | 1296 |
| 3 | 328 | 321 | 649 |
| 4 | 165 | 164 | 329 |
| 5 | 81 | 64 | 145 |
| 6 | 41 | 27 | 68 |
| 7 | 31 | 17 | 48 |
| 8 | 15 | 4 | 19 |
| 9 | 9 | 5 | 14 |
| 10 | 3 | 0 | 3 |
| 11 | 2 | 0 | 2 |
| 12 | 1 | 1 | 2 |
| 13 | 0 | 0 | 0 |
| 14 | 0 | 0 | 0 |
| 15 | 1 | 0 | 1 |
| 16 | 1 | 0 | 1 |

The principle behind this kind of tests is defining a statistic that serves to measure the distance between a theoretical distribution function $F_0(k)$ and the empirical (cumulative) distribution function for $n$ events, $F_n(k)$. Every value of the statistic is associated with a $p$-value, that can be interpreted as the probability of obtaining a value of the statistic at least as large as the one obtained, given that the null hypothesis.

$$\mathcal{H}_0 : F_n(k) = F_0 \tag{15}$$

is true. If the $p$-value is smaller than a previously defined threshold value $\alpha$, the null hypothesis is rejected.

Two separate tests were applied on our data:

1. A test of whether the observed data comes from a geometric distribution with $p = q = 0.5$. Based on the model outlined in section An 'Efficient Market' toy model for the distribution

**Table 6. Descriptive statistics of data presented in Tables 2–5.**

| Market | Mean | RMS | Skewness | Kurtosis |
|---|---|---|---|---|
| DJIA overall | 1.9713 ± 0.0185 | 1.3511 ± 0.0130 | 2.0215 ± 0.0334 | 8.9454 ± 0.9998 |
| DJIA uptrends | 2.0595 ± 0.0275 | 1.4298 ± 0.0194 | 1.9844 ± 0.0471 | 8.7538 ± 0.9996 |
| DJIA downtrends | 1.8375 ± 0.0229 | 1.1893 ±0.0162 | 1.8010 ± 0.0472 | 6.7113 ± 0.9996 |
| Nasdaq Overall | 2.2040 ± 0.0243 | 1.6825 ± 0.0172 | 2.3962 ± 0.0354 | 12.5405 ±0.9998 |
| Nasdaq uptrends | 2.4519 ± 0.0385 | 1.8884 ± 0.0273 | 2.1989 ±0.0500 | 10.9085 ± 0.9996 |
| Nasdaq downtrends | 1.9233 ± 0.0273 | 1.3378 ± 0.0193 | 2.2818 ± 0.0499 | 12.0090 ±0.9996 |
| IPC overall | 2.3653 ± 0.0276 | 1.8307 ± 0.0195 | 2.5248 ± 0.0369 | 14.8844 ± 0.9998 |
| IPC uptrend | 2.4899 ±0.0407 | 1.9202 ± 0.0288 | 2.0464± 0.0519 | 8.8614 ± 0.9996 |
| IPC downtrends | 2.1800 ± 0.0340 | 1.5995 ± 0.0240 | 2.4244 ± 0.0520 | 13.7515 ± 0.9996 |
| Nikkei overall | 1.9932 ± 0.0198 | 1.4214 ± 0.0140 | 2.2929 ±0.0341 | 11.3466 ± 0.9998 |
| Nikkei uptrends | 2.0735 ± 0.0304 | 1.5434± 0.0215 | 2.4070 ± 0.0481 | 12.0543 ± 0.9996 |
| Nikkei downtrends | 1.9094 ±0.0251 | 1.2774 ± 0.0178 | 1.9697 ±0.0482 | 8.4111 ±0.9996 |

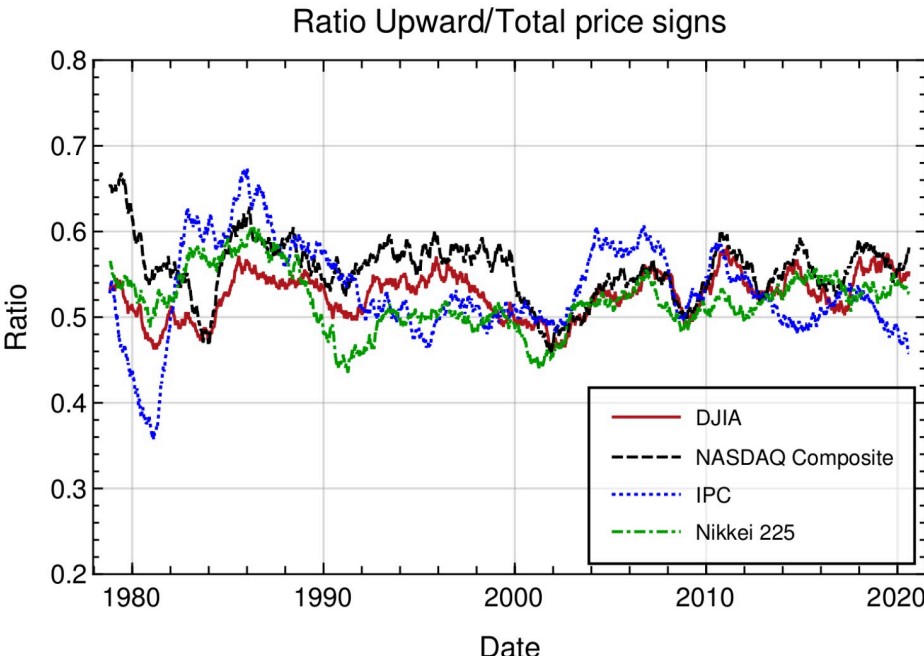

**Fig 3. Ratio of upward to total price changes in daily data, plotted against time for the interval from 10–30-1978 to 08–07-2020, calculated over a time window of 504 trading days.**

of run durations and empirical evidence on data, we interpret a rejection of this null hypothesis as evidence that the market is moving up or down in the investigated period.

2. A test of whether the observed data is drawn from a distribution belonging to a parametric family $\mathcal{G}(p)$. This tell us whether the up and down ticks can be modeled as a Bernoulli process.

These two tests will allow us to assess the validity of the Bernoulli hypothesis.

For a more complete discussion on the Anderson-Darling test for discrete data, including some comments about how it was applied to the geometric distribution case, see S1 Appendix.

## Data analysis

In order to motivate our analysis, in Fig 3, for the four markets studied in this paper, we show the ratio of the upward to total price changes in daily data plotted against time for the years 1978—2020. This ratio is calculated over an overlapping time window of 504 trading days shifted every 5 days. It can be seen that variations of this ratio fluctuate closer to the value of $\frac{1}{2}$ for DJIA and Nikkei, whereas for Nasdaq and IPC they are greater than those expected for the same time windows in a Bernoulli process with parameter $p = 1/2$.

Hereunder, we present different detailed studies on the four markets to see whether consecutive price increments/decrements with the same sign do follow a geometric distribution. Firstly, after estimating the distributions of the duration of uninterrupted uptrends and downtrends for all data, we separately and independently fit a geometric distribution to each one of these distributions, where the observed sum of the same duration uptrends and downtrends is the only constraint. For this reason, although we denote the estimated parameter $p$ and $q$ for uptrend and downtrend durations respectively, this is only nominal, since we fit those

parameters separately and independently without constricting the geometric fits to comply the constraint $p + q = 1$, i.e. we consider and analyze the sequences of uninterrupted uptrends and downtrends durations separately. Due to this reason, we are sometimes prone in this paper to refer only to the parameter $p$ in our discussion. More on this point can be found at the end of the current section.

Analyzed empirical data can be consulted in Tables 2–5, and their corresponding geometric fits are displayed in Fig 4 for all probability distributions of uptrend and downtrend durations corresponding to the four different indices studied here. A Maximum Likelihood Fit (MLF) was applied. The results of these fits can be consulted in Table 7. In Fig 4, black solid small circles represent observations, the geometric fit corresponds to the red solid line and, as a visual guide, blue dashed lines indicate a geometric distribution with parameter $p = q = 0.5$.

In order to obtain a good fit, with appropriate and correct $p$ and $\chi^2$ values, the fitting procedure was applied on the plots region where no null event gaps were observed in the trends duration distributions, i.e. the region where trend duration showed zero events for first time were excluded from the fit. Cut off applied are also shown in all corresponding plots of Fig 4 and are indicated by a dotted, vertical line. The only distribution that does not present any empty value in trend duration is the one corresponding to the DJIA uptrends and therefore it was fitted in the whole range of observed values.

From Fig 4 and Table 7, it can be seen that, although all discussed markets display some extreme trends durations deviating in different grades from the geometric model, for the whole of our data sample, distributions of increasing and decreasing trends durations for DJIA and Nikkei can be fitted reasonably by a geometric distribution with $p = \frac{1}{2}$, while the corresponding empirical runs distributions of Nasdaq and IPC are also reasonably fitted by a geometric distribution with parameters not necessarily equal to $\frac{1}{2}$. More on this facts will be discussed below and in next sections. From above fits, we can rank markets in order of decreasing distance from the $p = 0.5$ model, with Nikkei being the closest, followed by the DJIA then the Nasdaq, and the IPC being the most distant during the analyzed time period.

Finally, although for small and medium size trends, especially for the more mature markets DJIA and Nikkei, the geometric model with $p = \frac{1}{2}$ is a good approximation, it is not possible to conclude that the conditions $p = q = \frac{1}{2}$ and $p + q = 1$ are fulfilled in general for all market and every run duration. Classical analyses [24] reported periods where for runs of the monthly index DJIA, $p = 0.57$ and $q = 0.43$ (1897–1929) and in contrast the S&P monthly composite index is not even consistent with the equal probability condition $p = q = \frac{1}{2}$ and even with the probability conservation condition $p + q = 1$, for example for the period January 1871 to December 1917, for this index $p = 0.67$ and $q = 0.50$ and for the time span January 1918 to March 1956, $p = 0.6$ and $q = 0.60$. These results suggest that for these cases and the financial indices examined in [24], we are not dealing with a random process with $p = q = \frac{1}{2}$. In the classical reference [24], $p$ and $q$ values are not estimated by a fit procedure as the performed here: instead they calculate the relative frequencies of indices up and down events to different time scales.

More deviations observed empirically of the hypothesis $p = q = \frac{1}{2}$ are reported in [25–27]. For an interesting and more modern analysis on runs for high frequency financial data, see [28].

## Time variation of $p$ and $q$ and other estimates of these parameters

In order to gain a better insight about how our empirical trend duration distributions dynamically differ from the geometric theoretical distribution, the evolution in time of $p$ and $q$ values are plotted in the upper and lower left panels of Fig 5 respectively. These parameters are

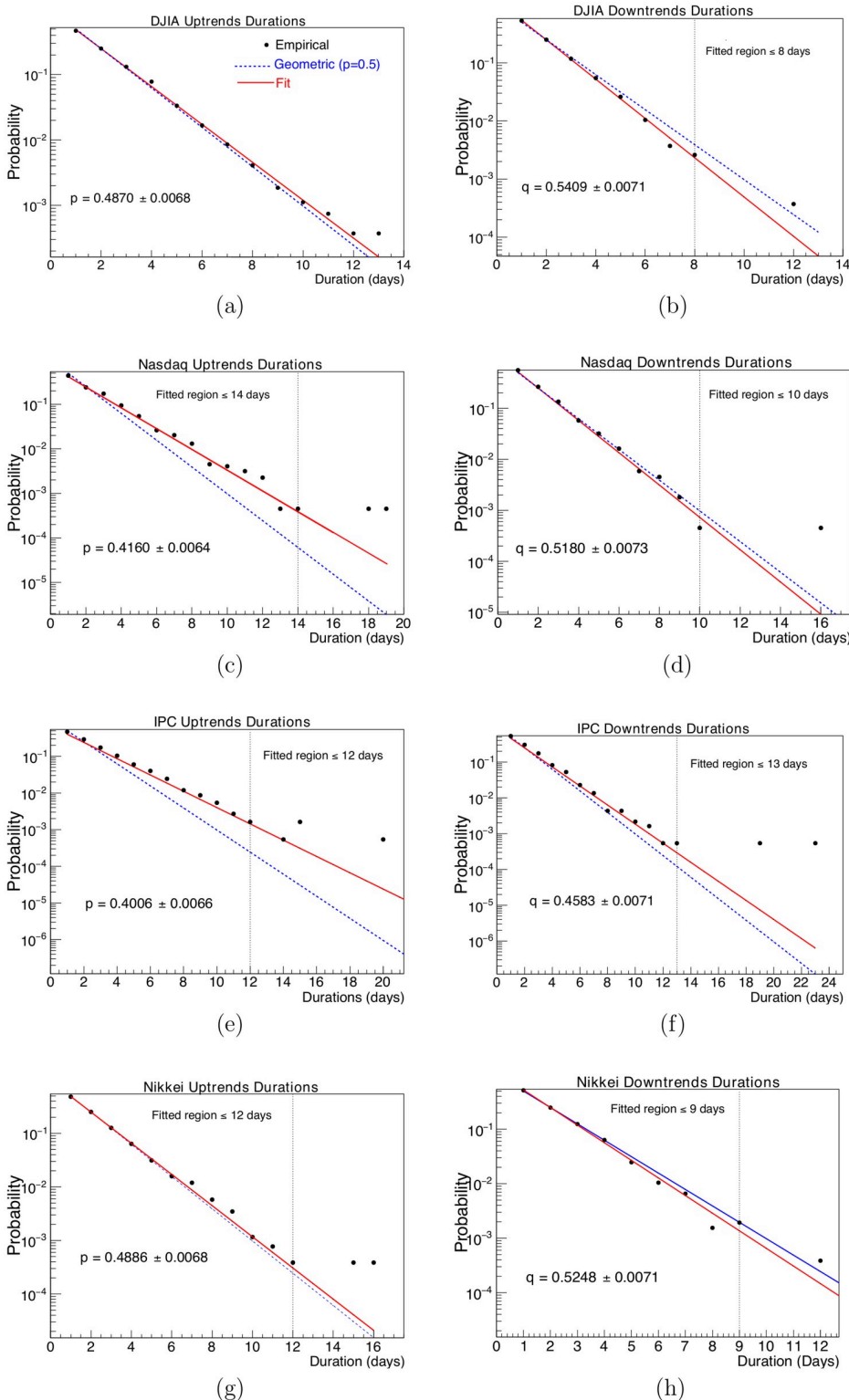

**Fig 4. Subfigures 4(a), 4(c), 4(e) and 4(g) present the uptrends duration distributions and subfigures 4(b), 4(d), 4(f) and 4(h) correspond to downtrends duration distributions.** The solid lines are the fitted geometric distributions while the dotted lines are the expected geometric distributions ($p = 0.5$). Parameters $p$ and $q$ are shown on each subfigure. Fit results are displayed in Table 7. Fitted regions are indicated in each plot by vertical dotted lines.

**Table 7. Fitted *p* and *q* parameters of the geometric model.** DJIA and Nikkei empirical trend durations distribution are well fitted by the geometric model. Nasdaq and IPC are not. NDF means "number of degrees of freedom". Fits were performed on the data listed in Tables 2–5.

| Market | Fitted region uptrends (days) | $p$ | $\chi^2/NDF$ uptrends | Fitted region downtrends (days) | $q$ | $\chi^2/NDF$ downtrends |
|---|---|---|---|---|---|---|
| DJIA | $\leq 13$ | 0.4870±0.0068 | 8.5436/12 | $\leq 8$ | 0.5409±0.0071 | 2.6636/7 |
| Nasdaq | $\leq 14$ | 0.4160±0.0064 | 15.9685/13 | $\leq 10$ | 0.5180±0.0073 | 3.8665/9 |
| IPC | $\leq 12$ | 0.40006±0.0066 | 2.0455/11 | $\leq 13$ | 0.4583±0.0071 | 8.5935/12 |
| Nikkei | $\leq 12$ | 0.4886±0.0068 | 6.6170/11 | $\leq 9$ | 0.5248±0.0071 | 7.0781/8 |

independently calculated over a time window of 252 trading days shifted every ten days, separately over the sequences of observed uninterrupted uptrends and downtrends durations. We see that their corresponding values tend to oscillate around $\frac{1}{2}$ and that markets with *p* and *q* values closer to $\frac{1}{2}$ are DJIA and Nikkei. Empirical distributions of the calculated values of *p* and *q* for all markets are displayed in the right, upper and lower panels of same Fig 5 respectively. The corresponding mean and standard deviation of *p* and *q* values are displayed in Table 8,

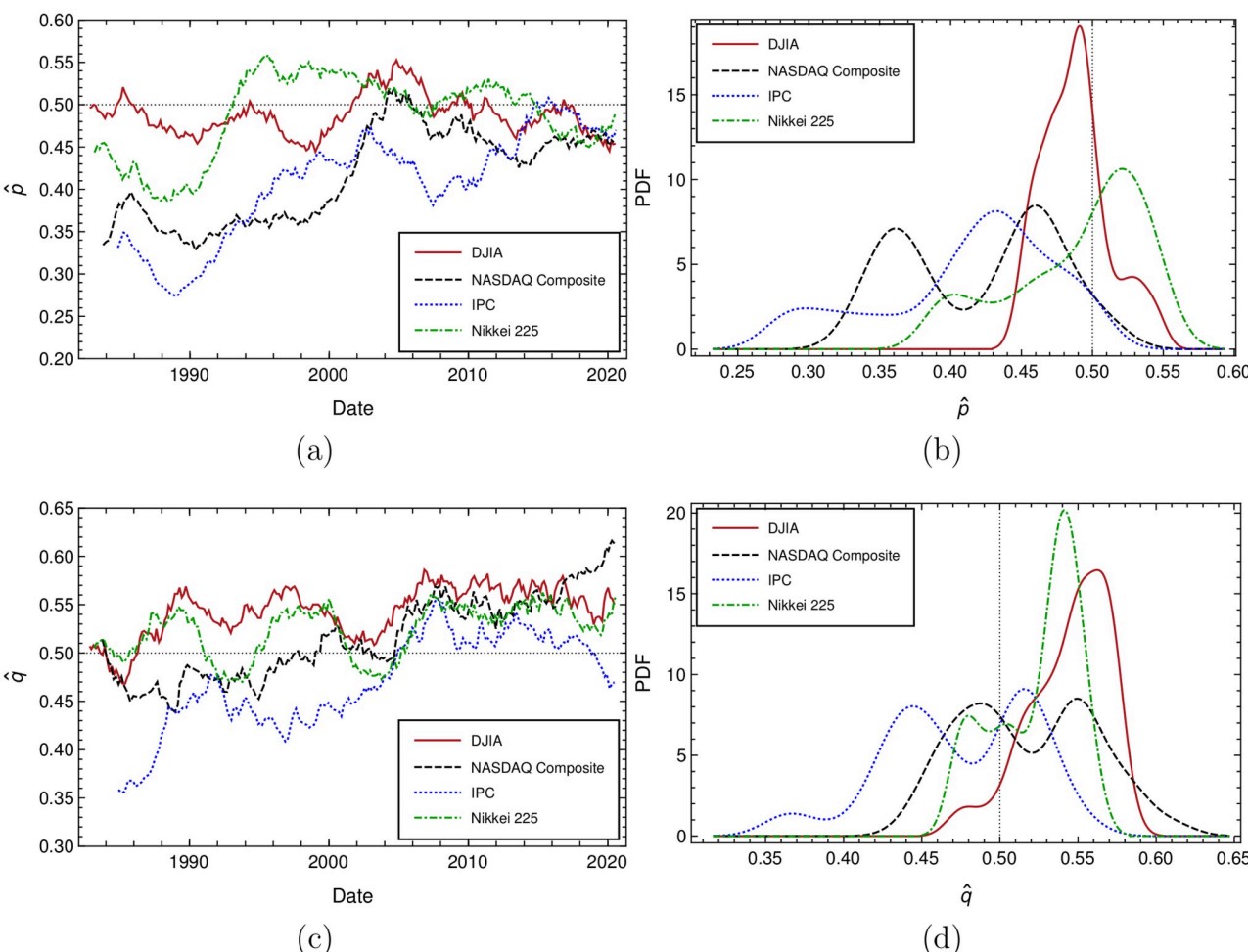

**Fig 5.** 5(a): *p* variation over time calculated for all data samples with a time window of 252 days shifted each 10 trading days. 5(b): Empirical distributions of *p* from left subfigure. 5(c): *q* evolution in time for all analyzed data samples. 5(d): Empirical distributions of *q* from left subfigure.

**Table 8. Mean and standard deviation values of $p$ and $q$ distributions shown in Fig 5, and generated with a rolling, overlapping time window of 252 days.** Same values generated for two additional overlapping time frames of 200 and 300 days are also displayed.

| Market | Time window (overlapping) | $<p>$ | $\sigma_p$ | $<q>$ | $\sigma_q$ |
|---|---|---|---|---|---|
| DJIA | 200 | 0.4877±0.0017 | 0.0266±0.0012 | 0.5437±0.0017 | 0.0276±0.0012 |
| | 252 | 0.4881±0.0015 | 0.0236±0.0011 | 0.5440±0.0016 | 0.0257±0.0012 |
| | 300 | 0.4881±0.0014 | 0.0220±0.0010 | 0.5445±0.0016 | 0.0240±0.0011 |
| Nasdaq | 200 | 0.4210±0.0038 | 0.0561±0.0027 | 0.5198±0.0030 | 0.0448±0.0021 |
| | 252 | 0.4208±0.0037 | 0.0543±0.0026 | 0.5187±0.0029 | 0.0427±0.0021 |
| | 300 | 0.4207±0.0037 | 0.0534±0.0026 | 0.5177±0.0028 | 0.0406±0.0020 |
| IPC | 200 | 0.4130±0.0044 | 0.0629±0.0031 | 0.4732±0.0036 | 0.0506±0.0025 |
| | 252 | 0.4129±0.0044 | 0.0616±0.0031 | 0.4741±0.0034 | 0.0472±0.0024 |
| | 300 | 0.4128±0.0043 | 0.0600±0.0030 | 0.4749±0.0032 | 0.0450±0.0023 |
| Nikkei | 200 | 0.4892±0.0031 | 0.0486±0.0022 | 0.5235±0.0019 | 0.0295±0.0014 |
| | 252 | 0.4895±0.0031 | 0.0473±0.0022 | 0.5233±0.0017 | 0.0264±0.0012 |
| | 300 | 0.4898±0.0031 | 0.0463±0.0022 | 0.5234±0.0016 | 0.0243±0.0011 |

where other size rolling time windows were also used in their calculation. It can be verified that listed values are all consistent with those estimated by the geometric fit procedure shown in Fig 4 with estimated fit parameters given in Table 7.

Here it is important to mention the estimate of $p$ and $q$ shown in Table 8, at this moment only serves to corroborate the values obtained by the geometric fitting procedure. We mention this, because notwithstanding the independence of both measurements and the agreement between values displayed in both Tables 7 and 8, entries of distributions shown in the two right histograms in Fig 5 are not all really statistically independent, since they were calculated by using a rolling time window of 252 days as described above. In addition to this, $p$ and $q$ values show a certain degree of non-stationarity and finally, these results can be dependent on the choice of the time-window size. Even considering these three facts, the agreement between the estimates obtained by these two different procedures over different time frames is remarkable. Taking in consideration these facts and in order to confirm the quality of our estimation, we show in Table 9 the results obtained, this time using no overlapping time windows of again 200, 252 and 300 days.

**Table 9. Mean values and standard deviation of $p$ and $q$ distributions, generated this time by using no overlapping time windows, of of 200, 252 and 300 days.** Obtained values are consistent with those shown in previous Table 8.

| Market | Time window (no overlapping) | $<p>$ | $\sigma_p$ | $<q>$ | $\sigma_q$ |
|---|---|---|---|---|---|
| DJIA | 200 | 0.4874±0.0082 | 0.0297±0.0061 | 0.5436±0.0082 | 0.0295±0.0060 |
| | 252 | 0.4906±0.0076 | 0.0240±0.0057 | 0.5438±0.0074 | 0.0234±0.0055 |
| | 300 | 0.4867±0.0067 | 0.0231±0.0058 | 0.5427±0.0091 | 0.0257±0.0069 |
| Nasdaq | 200 | 0.4161±0.0172 | 0.0596±0.0127 | 0.5158±0.0122 | 0.0404±0.0090 |
| | 252 | 0.4148±0.0205 | 0.0616±0.0154 | 0.5192±0.0152 | 0.0457±0.0114 |
| | 300 | 0.4155±0.0207 | 0.0584±0.0156 | 0.5101±0.0133 | 0.0353±0.0102 |
| IPC | 200 | 0.4124±0.0200 | 0.0663±0.0148 | 0.4661±0.0179 | 0.0594±0.0133 |
| | 252 | 0.4047±0.0228 | 0.0646±0.0173 | 0.4665±0.0204 | 0.0577±0.0154 |
| | 300 | 0.4080±0.0252 | 0.0667±0.0193 | 0.4691±0.0237 | 0.0628±0.0181 |
| Nikkei | 200 | 0.4875±0.0145 | 0.0502±0.0107 | 0.5240±0.0091 | 0.0315±0.0091 |
| | 252 | 0.4870±0.0160 | 0.0507±0.0120 | 0.5243±0.0101 | 0.0320±0.0075 |
| | 300 | 0.4874±0.0178 | 0.0504±0.0135 | 0.5234±0.0095 | 0.0269±0.0072 |

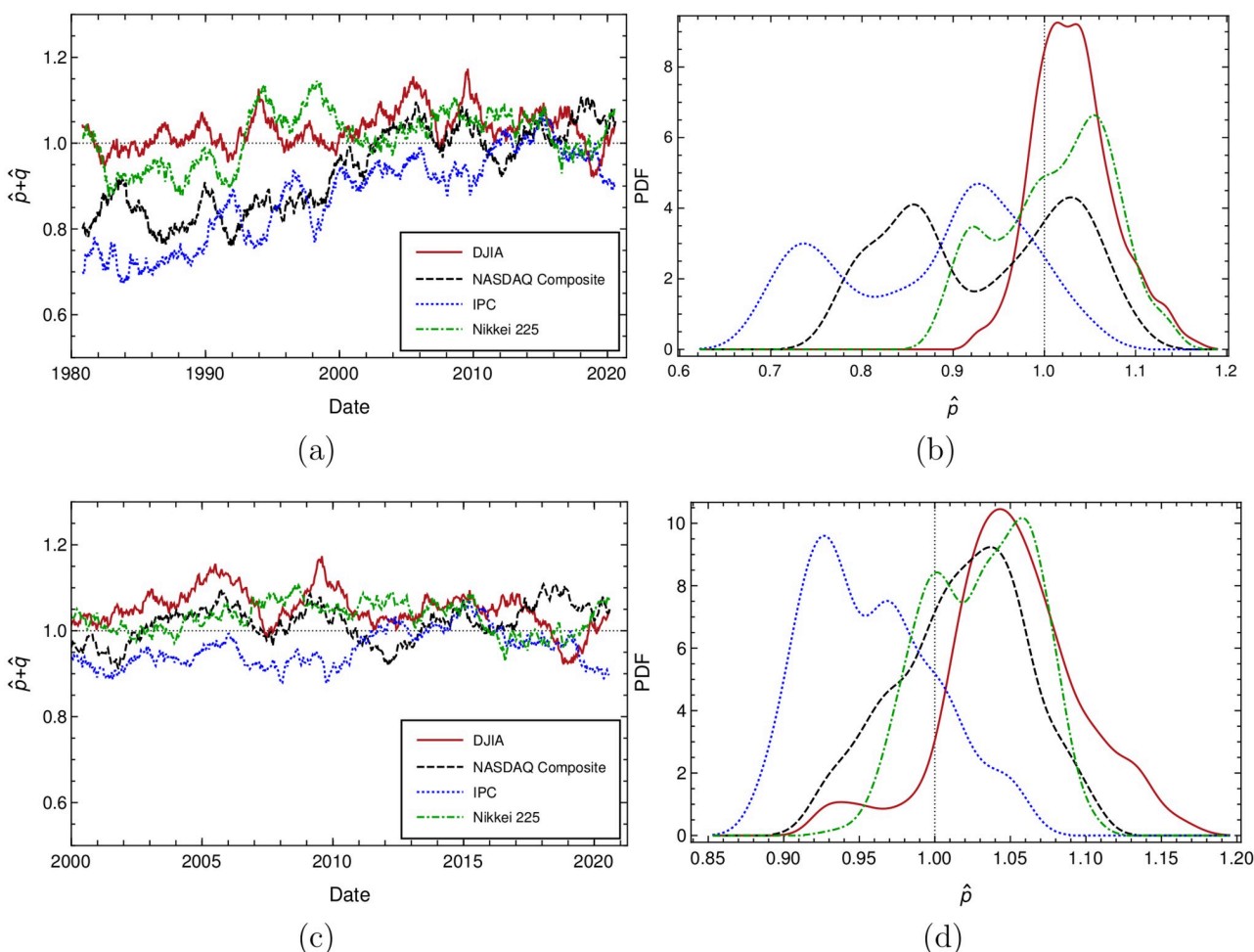

**Fig 6.** 6(a) is $p + q$ vs time; 6(b) is $p + q$ distribution for studied data; 6(c) is $p + q$ vs time after 2000 year; and, 6(d) is $p + q$ distribution after year 2000.

To end this subsection, we observe that the distance from the $p = 0.5$ model for the different studied markets established at the end of section Data analysis by the geometric fitting procedure is again confirmed by the values of $p$ and $q$ showed in Table 9.

## Mean value and variance of $p + q$ distribution

Even if we calculate $p$ and $q$ independently and we use this notation in nominal way, in this subsection and for completeness reasons, we carefully study the probability conservation that a geometric stochastic process must to meet, i.e. $p + q = 1$; in order to see what is happening, we show in Fig 6(a), the behavior of $p + q$ as a function of time, calculated as explained before, by using a 252 trading days rolling time frame shifted each 10 days. It can be seen that $p + q$ for DJIA and Nikkei oscillates around 1, whereas IPC and Nasdaq get closer on time to this value and, then, after year 2000, they follow the same behavior than DJIA and Nikkei. The upper right panel of same figure shows $p + q$ empirical distribution for all studied markets. Non stationarity effects are observed in all of them in different degree, however mean value of $p + q$ are close to 1 in all those distributions. Cutting off all data previous to year 2000 and repeating

**Table 10. Mean and standard deviation of $p + q$ distributions, where rolling, overlapping time frames of 200, 252 and 300 trading days were set up and shifted each 10 days.** Observed $p + q$ mean values are very close to the value of 1 for Nikkei and DJIA and despite non stationarity, are also close to 1 for Nasdaq and IPC markets. Restricting our analysis to dates thereafter year 1999, clearly $p + q$ values are even nearest to the value of 1 for all markets.

| Market | Time window (overlapping) | $<p + q>$ | $\sigma_{(p + q)}$ | $<p + q>$ after 2000 | $\sigma_{(p + q)}$ after 2000 |
|---|---|---|---|---|---|
| DJIA | 200 | 1.0394±0.0020 | 0.0660±0.0014 | 1.0600±0.0038 | 0.0705±0.0022 |
| | 252 | 1.0321±0.0018 | 0.0435±0.0010 | 1.0513±0.0020 | 0.0451±0.0014 |
| | 300 | 1.0357±0.0017 | 0.0055±0.0013 | 1.0559±0.0030 | 0.0584±0.0018 |
| Nasdaq | 200 | 0.9425±0.0034 | 0.1080±0.0024 | 1.0267±0.0027 | 0.0612±0.0019 |
| | 252 | 0.9340±0.0030 | 0.1046±0.0023 | 1.0180±0.0019 | 0.0424±0.0013 |
| | 300 | 0.9378±0.0032 | 0.1023±0.0023 | 1.0223±0.0023 | 0.0521±0.0016 |
| IPC | 200 | 0.8830±0.0035 | 0.1133±0.0025 | 0.9621±0.0028 | 0.0633±0.0020 |
| | 252 | 0.8750±0.0033 | 0.1092±0.0024 | 0.9680±0.0019 | 0.0425±0.0013 |
| | 300 | 0.8792±0.0033 | 0.1065±0.0024 | 0.9592±0.0023 | 0.0532±0.0023 |
| Nikkei | 200 | 1.0199±0.0025 | 0.0790±0.0018 | 1.0390±0.0024 | 0.0556±0.0017 |
| | 252 | 1.0124±0.0020 | 0.0620±0.0014 | 1.0306±0.0016 | 0.0350±0.0011 |
| | 300 | 1.0163±0.0022 | 0.0711±0.0016 | 1.0359±0.0020 | 0.0456±0.0014 |

this analysis, it is observed that indeed $p + q$ for all markets fluctuate closer and around the value of 1, and that even runs of IPC and Nasdaq markets turn closer to geometric as time passes. Distributions of these fluctuations are plotted in Fig 6(d). Corresponding mean and standard deviation of $p + q$ distributions for all analyzed and restricted after year 1999 data, can be seen in Table 10, where also we have calculated these mean values for 200 and 300 trading days rolling time windows.

In the same way we proceeded in previous subsection Time variation of $p$ and $q$ and other estimates of these parameters, we calculate mean values and RMS of $p + q$ for no overlapping time frames of 200, 252 and 300 days. Obtained values are shown in below Table 11, also calculated for all time period of the recorded data sample and for the span of time after year 2000.

For the full period of all analyzed data and from the measurements shown in Tables 10 and 11, we can rank studied markets in the following order of closeness to the geometric distribution: 1) DJIA, 2) Nikkei, 3) Nasdaq and lastly 4) IPC.

**Table 11. Again, mean and standard deviation of $p + q$ distributions, this time calculated by using 200, 252 and 300 no overlapping time frames.** Displayed values are consistent with those of Table 10.

| Market | Time window (no overlapping) | $<p + q>$ | $\sigma_{(p + q)}$ | $<p + q>$ after 2000 | $\sigma_{(p + q)}$ after 2000 |
|---|---|---|---|---|---|
| DJIA | 200 | 1.0394±0.0094 | 0.0678±0.0067 | 1.0597±0.0147 | 0.0750±0.0106 |
| | 252 | 1.0355±0.0094 | 0.0603±0.0067 | 1.0574±0.0140 | 0.0451±0.0014 |
| | 300 | 1.0369±0.0097 | 0.0573±0.0069 | 1.0571±0.0140 | 0.0596±0.0102 |
| Nasdaq | 200 | 0.9386±0.0150 | 0.1081±0.0150 | 1.0249±0.0107 | 0.0546±0.0077 |
| | 252 | 0.9394±0.0170 | 0.1087±0.0122 | 1.0238±0.0139 | 0.0424±0.0013 |
| | 300 | 0.9366±0.0170 | 0.1004±0.0121 | 1.0216±0.0113 | 0.0479±0.0082 |
| IPC | 200 | 0.8806±0.0163 | 0.1175±0.0116 | 0.9599±0.0143 | 0.0727±0.0103 |
| | 252 | 0.8773±0.0184 | 0.1179±0.0131 | 0.9604±0.0129 | 0.0580±0.0129 |
| | 300 | 0.8743±0.0188 | 0.1095±0.0134 | 0.9573±0.0120 | 0.0496±0.0088 |
| Nikkei | 200 | 1.0164±0.0108 | 0.0771±0.0077 | 1.0352±0.0120 | 0.0601±0.0087 |
| | 252 | 1.0182±0.0121 | 0.0769±0.0087 | 1.0404±0.0108 | 0.0483±0.0078 |
| | 300 | 1.0149±0.0129 | 0.0751±0.0092 | 1.0351±0.0129 | 0.0534±0.0094 |

**Estimation of $<p>$ and $\sigma_p$.** We have estimated $p$ by applying the usual, one parameter fitting procedure illustrated in section Data analysis. We have seen that the estimate value of $p$ for the different data samples, are compatible with the corresponding values of $<p>$ obtained by averaging data for each movable overlapping and not overlapping time windows with sizes given in Tables 8 and 9. The above and the following is, of course, also valid for the case of $q$.

The process of finding the maximum likelihood estimate of the parameter $p$ in a geometric distribution as given by Eq (14):

$$P(x) = (1 - p)^x p, \; x = 0, 1, \ldots \tag{16}$$

is a well known methodology which consists of finding the value $\hat{p}$, of $p$, which maximizes the likelihood function. For the case of the geometric distribution, given a random sample $x_1, \ldots, x_n$, we obtain:

$$\hat{p} = \frac{1}{1 + \bar{x}} \tag{17}$$

where $\bar{x}$ denotes the sample mean.

From the asymptotic properties of the MLE estimators, see [29, 30], $\hat{p}$ has approximately a normal distribution with mean $<p>$ and variance $n^{-1} p^2 (1 - p)$. Formulas to calculating the error of the mean and variance are well known, see [31], although their estimation is usually automatically made in the background by the scientific software used to perform the data analysis, in our case Mathematica.

To conclude this subsection, we must point out that the measurements show that $p + q$ is slightly greater than 1 for DJIA and Nikkei, are not contradictory with empirical experience, since by studying runs for different time scales, a slight excess of uptrends over downtrends has been observed in financial data at least since the 1930s [24–26]. Remember also that our two measurements of $p$ and $q$ were performed in an independent way and that the early financial literature also evidences that, at least for some time spans, the evolution of runs is not well represented by a random walk with equal probabilities of going up and down. We believe these empirical facts are well known in financial econometrics, but may not be well-known by physicists.

In this paper, we do not only confirm these experimental facts and show time evolution of $p$, $q$ and $p + q$, but in next section, we will estimate the fraction of time runs of markets follow a geometric behavior with $p = \frac{1}{2}$ and with any $p$.

## Anderson-Darling test in the case $p = 0.5$

To study dynamically how the theoretical statistical model differs from the empirical data, we calculate the Anderson-Darling statistics for the corresponding trends durations of the observed empirical distribution and the theoretical, geometric distribution with parameter $p = 0.5$. Fig 7 display the obtained $p$-values of the Anderson-Darling statistic, $A_n^2$ for different time periods (not to be confused with the $p$ parameter of the geometric distribution). Remember that in the case we are interested, a $p$-value is the probability of obtaining a value of $A_n^2$ at least as big as the one that was really obtained, given that the probability distribution is actually geometric.

Analysis presented in Fig 7(a) shows that for the DJIA, the greatest deviations from the geometric distribution with parameter $p = \frac{1}{2}$, occurred between the years 2002–2011. Fig 7(b) for Nasdaq, it is observed that as time goes by, $p$-values of the Anderson-Darling show that empirical data tends to agree better with the geometric distribution $\mathcal{G}(0.5)$, especially after year 2000. Fig 7(c) shows that, similar to NASDAQ, the IPC index agreement between data and the

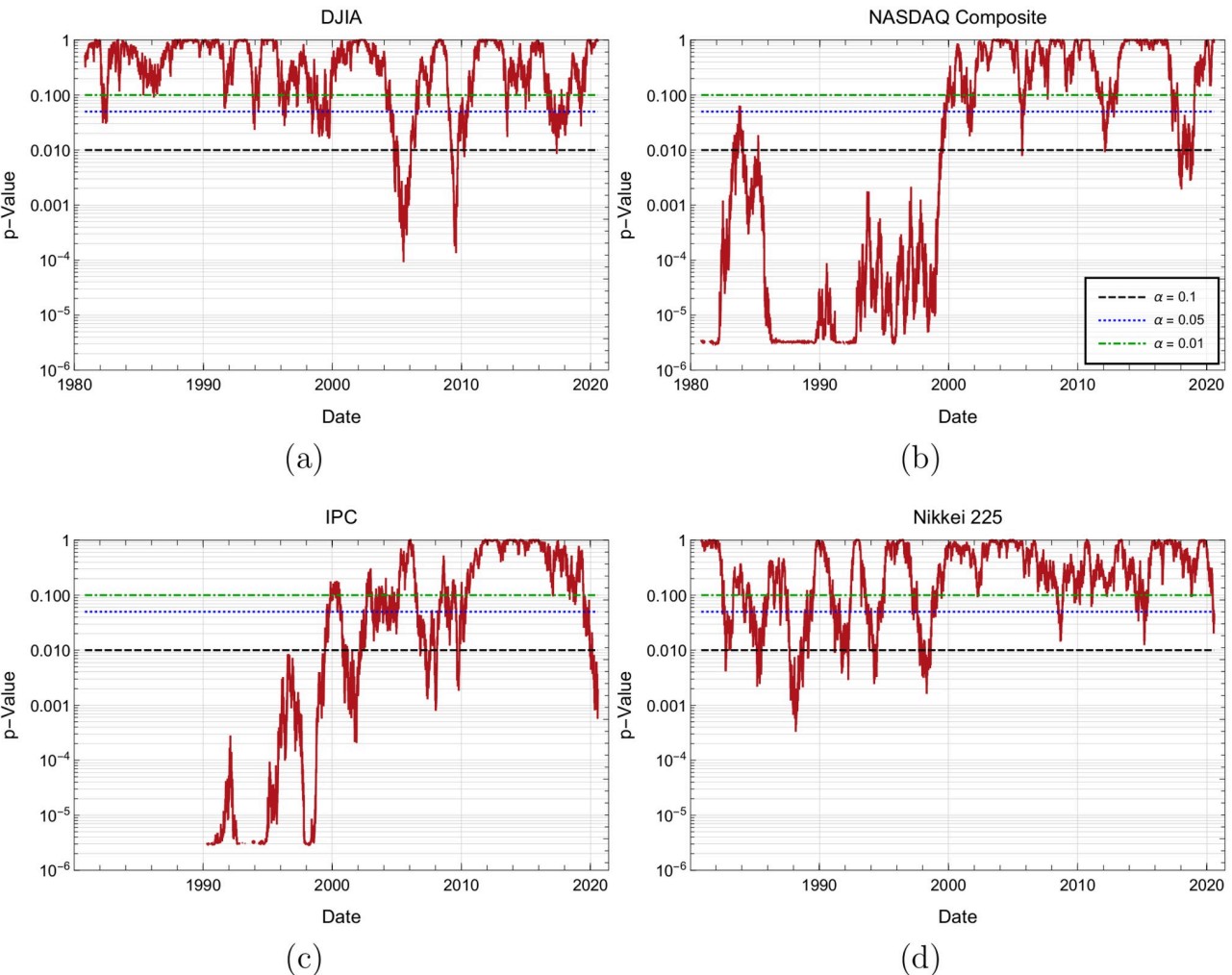

**Fig 7. Subfigures 7(a) to 7(d) all show the *p*-values of the Anderson-Darling statistic evolution on time for DJIA, Nasdaq, IPC and Nikkei indices, respectively.**

geometric distribution increases with time. Finally, Fig 7(d) shows that for the Nikkei case, *p*-values of the Anderson-Darling show a good agreement between the Geometric model with $p = \frac{1}{2}$ and the observed trend duration distribution.

As an auxiliary analysis, Fig 8 shows the dates when the events from Fig 7 have a *p*-value below the $\alpha = 0.05$ significance level, or in other words, the dates for which the null hypothesis can be rejected, with a significance level $\alpha = 0.05$ and the complementary dates for which the geometric hypothesis with $p = q = \frac{1}{2}$ cannot be rejected.

The above observations are compatible with the plot shown in Fig 5 presented in subsection Time variation of *p* and *q* and other estimates of these parameters, where it is shown that the greatest, however diminishing deviations from the geometric distribution with $p = 0.5$ occurred between the years 1980–2000, especially for Nasdaq and IPC and to a lesser extent for DJIA and Nikkei. In next subsection Anderson-Darling parametric test for the geometric distribution, we shall show that, in all cases, still the geometric model upholds, by allowing the parameter *p* to vary freely.

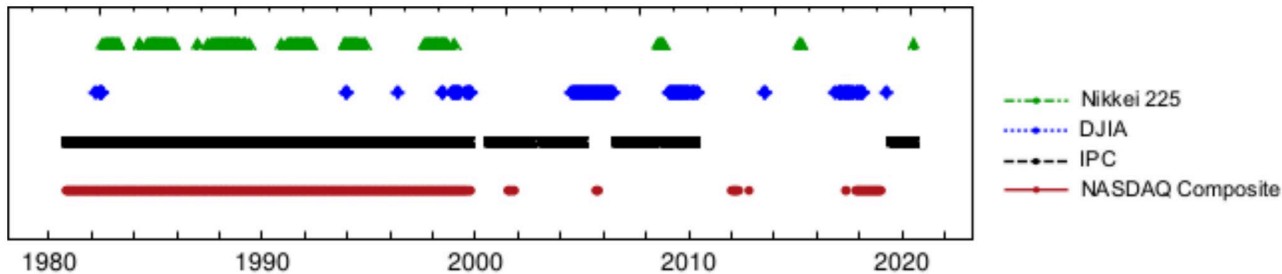

**Fig 8. Colored points represent the dates where events from Fig 7, have a *p*-value below the *α* = 0.05 significance level, i.e. dates where the statistical behavior of the durations of the uninterrupted trends are more distant from a geometric with parameter *p* = 0.5.** For easy reading colored points are enlarged.

### Anderson-Darling parametric test for the geometric distribution

Let us explore the possibility that trend durations follow a geometric distribution with any parameter $p \in (0, 1)$. Results of this parametric test are displayed in Fig 9. It can be observed that, for all studied markets, the assumption of a Bernoulli process for price directions holds reasonably well for most of the time, except for sporadic deviations that are usually related to extreme market movements such as in the case of a financial crisis. Again Fig 10 is an auxiliary figure that shows the dates when events from Fig 9 have a $p$-value below the $\alpha = 0.05$ significance level, i.e. the dates when studied markets do not follow the geometric model at the mentioned significance level. As it may be seen, for an important fraction of time, markets do seem to follow the geometrical model with some parameter $p$.

An application of previous results will be discussed in the next section A simple application: Assessing the fraction of time markets runs follow a geometric distribution.

### A simple application: Assessing the fraction of time markets runs follow a geometric distribution

Continuing the discussion at the end of section Data analysis, we can also use the results presented there to assess the percentage of time that the market follows the geometric model with $p = \frac{1}{2}$ and in a $p$ free parametric way. In order to do this we propose the following methodology:

1. Calculate a time series of $p$-values from the sample of trends durations using the geometric process with $p = 0.5$ as the null hypothesis.

2. Count the number of points above the significance level value $\alpha = 0.05$.

3. Divide it by the length of the time series to obtain the percentage of time the market has behaved as a market following the geometric process.

Repeat it for the case non parametric, i.e. the same null hypothesis but now with any $p$.

Following above criterion, we rank studied markets as follows: closest for a bigger time fraction to the geometric model with $p = \frac{1}{2}$ was the DJIA, followed by Nikkei 225, then the NASDAQ Composite and the end, the IPC. Here, under this criterion more mature markets runs follow closer the geometric model with $p = \frac{1}{2}$ for a longer, but this time Nikkei 225 and DJIA exchange rank position. Results obtained by means of this methodology, for the geometric case with $p = \frac{1}{2}$ as well as for a parametric free way, may be consulted in Table 12.

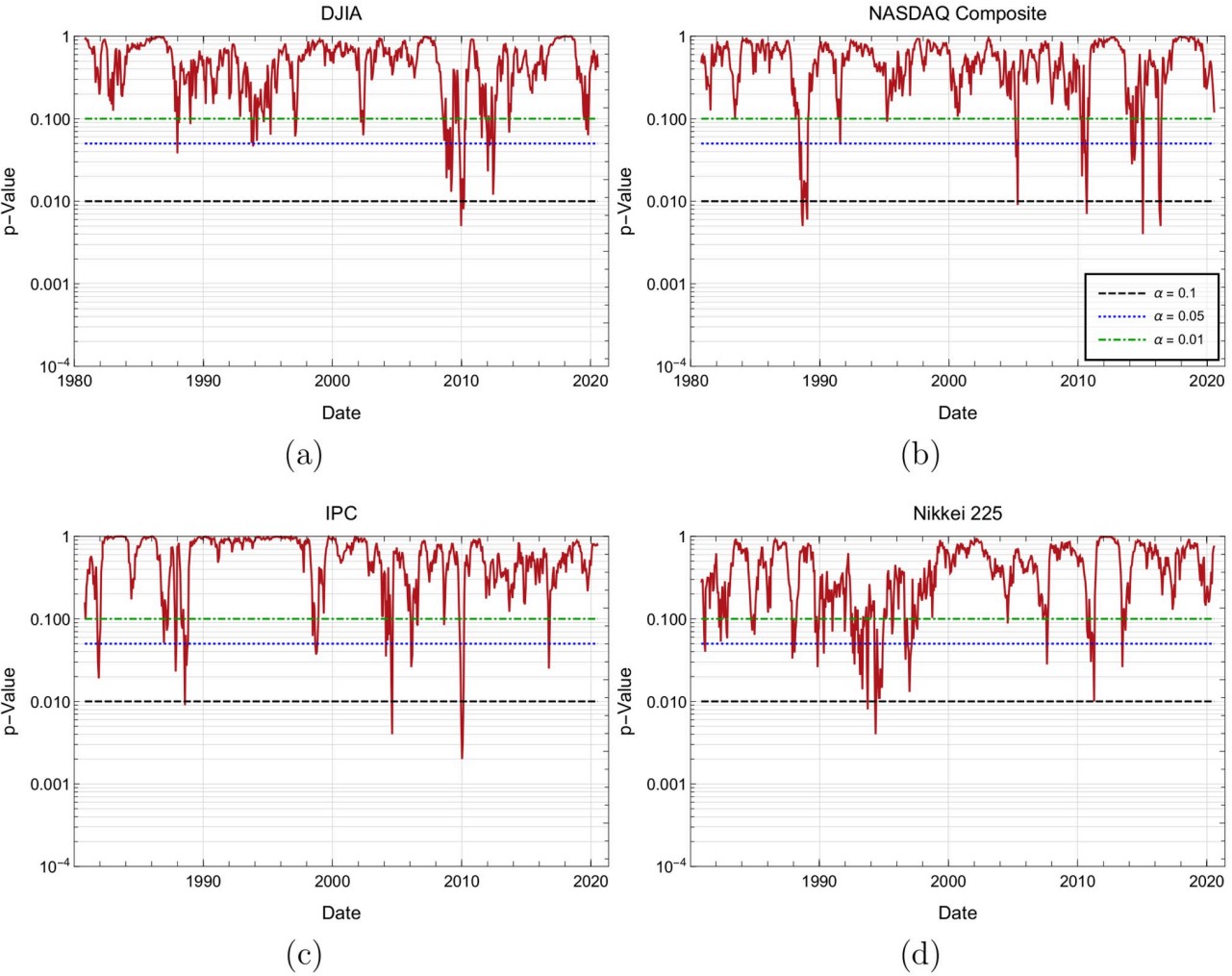

**Fig 9. _p_-values of the parametric family Anderson-Darling statistic for the studied markets.** 9(a) is DJIA, 9(b) is Nasdaq, 9(c) is IPC and 9(d) is Nikkei.

## Conclusions

The study of runs used to be an important research area in financial econometrics [24–27]. Some interesting more modern empirical studies have been performed on daily and high frequency data [28, 32, 33]; runs have been applied for example to assess market randomness [34] and even flash crashes in high frequency data [32]. In this paper, the probability distributions of the duration of elementary trends or price runs were studied for the market indices Dow Jones Industrial Average (DJIA), NASDAQ Composite, the Mexican Índice de Precios y Cotizaciones (IPC) and for the Japanese Nikkei 225. According to the discussion of section An 'Efficient Market' toy model for the distribution of run durations, these distributions are expected to be geometric, with parameter $p = 0.5$ and memoryless. Indeed the geometric distribution with $p = \frac{1}{2}$, provides a good model for trends of small and medium size, lets say until 10 or 12 days, for DJIA and Nikkei, and with $p$ non necessarily $\frac{1}{2}$ for less mature markets. On the other side, we show that trend duration distributions in all markets display outliers, however

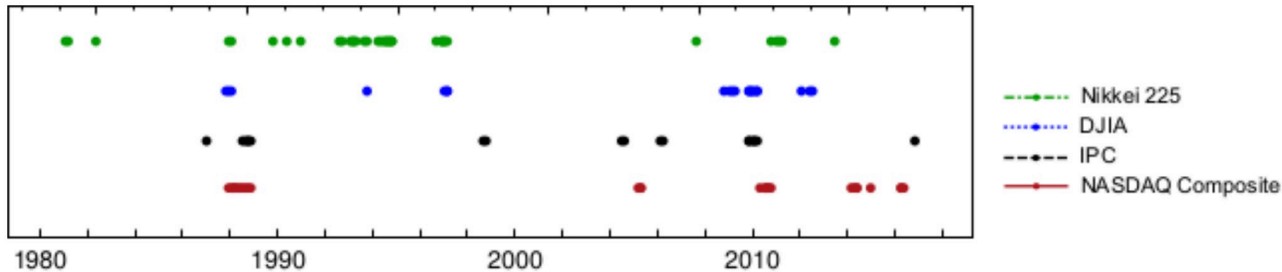

**Fig 10. Colored points, show dates from the parametric test where events observed in Fig 9 have a *p*-value below the *α* = 0.05 and then geometric model for any *p* can not be applied to describe runs size distribution for the different analyzed markets at that significance level.** Again, for easy reading colored points are enlarged.

more statistics is needed to study these extreme events that do not seem to follow the geometric model.

Additionally, by selecting overlapping and non overlapping time frames of 200, 252 and 300 trading days, we display $p$ and $q$ behavior over time, and the distribution of these parameters, allowing us to estimate their mean and RMS values and compare the former with the corresponding values obtained by a fitting procedure. Agreement obtained is remarkably good. We have shown that for all markets $p$ and $q$ values are evolving towards the value of $\frac{1}{2}$. The $p + q$ evolution over time is also displayed and by using same methodology we observe that $<p + q>$ is approaching over time to the value of one for all markets. Finally, markets with uninterrupted trends durations closer to follow a geometric behavior with $p = \frac{1}{2}$ may be ranked in the following order: Nikkei, DJIA, Nasdaq and IPC, meaning more mature market are closer to the geometric behavior with $p = \frac{1}{2}$.

Anderson-Darling test has been used to quantify the likelihood that a series of trends durations were generated by a process compatible with the geometric model with $p = \frac{1}{2}$; we also employed it to assess for how long, trends durations follow the geometric distribution with $p = \frac{1}{2}$, as well as for any other value of the parameter $p$. Corresponding dates during which markets runs do not follow the geometric behavior for $p = \frac{1}{2}$, and in a parametric free way, are displayed respectively in auxiliary Figs 8 and 10. Numerical time fractions displayed in Table 12 correspond to the fraction of the time markets follow the geometric distribution with $p = \frac{1}{2}$ and in parametric free way. First column of this table shows that for the significance level of 5%, price runs distribution of DJIA follows a geometric distribution with $p = \frac{1}{2}$ the 84% of the time, Nikkei 81%, Nasdaq 47% and IPC 37% of the time. Ranking obtained by this criterion, although exchange DJIA and Nikkei 225 positions, once again classifies more mature markets at the top of the list. Second column of same Table 12 shows that the distributions of all studied markets trends duration are close to a geometric distribution with a parameter $p$

**Table 12. Fraction of time, the overall of the studied data trends durations follow a geometric distribution with parameter *p* = 0.5, and with any *p*, both cases for a significance level of 5%.**

| Market/Case | Time fraction p = 0.5 | Time fraction Non Parametric |
|:---:|:---:|:---:|
| DJIA | 0.84 | 0.9673 |
| Nasdaq | 0.47 | 0.9626 |
| IPC | 0.37 | 0.9683 |
| Nikkei | 0.81 | 0.9419 |

not necessarily equal to $\frac{1}{2}$ a high fraction of all time. This fact can be supported by the quality of fits and respective fit parameters non equal to $\frac{1}{2}$ displayed in Fig 4 and Table 7, showing that with the exception of the few extreme values, geometric model fits well Nasdaq and IPC runs duration, with parameter $p$ (and $q$) non necessarily close to $\frac{1}{2}$.

Obtained results also show us that for more mature markets runs distribution are closer for a longer time spans to the geometric distribution with $p = \frac{1}{2}$, and that less mature markets runs seems to evolve on time to this same distribution. This empirical result reminds us the fact that worldwide markets increase their efficiency with time [35–38].

In section An 'Efficient Market' toy model for the distribution of run durations, we state that the geometric model with $p = \frac{1}{2}$ applied to price runs it may consistent with the EMH. However, if the empirical analysis falsifies the process, this does not mean that the EMH is falsified. In our opinion, more and deeper study should be necessary to clarify these facts, given that market efficiency refers to returns and not to price runs.

Finally, besides the above mentioned problem of making explicit the relation between market efficiency and the geometric behaviour of price runs, we have some additional remarks possibly leading to future work: in this paper, we analyzed regularly sampled data i.e, daily close price data, and although the geometric model seems well suited to model short and medium price trends durations, this observable is really a continuous random variable and conceivably the geometric model might not be suitable to describe non regularly sampled data [33], as for example in tick-by-tick data. The second remark has to do with the extreme values observed in the different trends durations distributions that occur with a higher probability than expected from the geometric model, as can be observed for values of trend durations above cut-off values signaled in the different panels of Fig 4 and recorded in the second and fifth columns of Table 7. In the present analysis, we observe at most two of these extreme events in the different panels of Fig 4, which is insufficient for saying something interesting on the distribution of these outliers. Also it will be interesting to study the relation of these runs extreme events with extreme returns events, particularly financial crashes; for example by using smaller time windows in our analyses. Third, although by their construction, in any data sample the number of downtrends and uptrends must be the same or their difference at most of one unit; from the composition of trends shown in Tables 2–5, it can be seem that for very short duration trends, number of downtrends predominate and for medium and long duration trends there are more uptrends than downtrends, this asymmetry and its relation with corresponding returns deserves a more detailed study.

In our opinion, and even if data analyses such as the one presented here have a long history, we have managed to find new results of possible interest to the econophysicist and financial communities. Specifically, apart from independent and consistently estimating the parameters $p$ and $q$, by two different methods, we show their time evolution as well as the time evolution of their addition $p + q$. Moreover, we not only show that the runs distribution of the markets studied is compatible with the geometric distribution with the estimated parameters, but we also estimate when and the fraction of the time during which the markets follow this behavior, parametrically for $p = q = 0.5$ and non-parametrically. The detection of when and for how long the distribution of the durations of market price runs have a geometric distribution is in our opinion our most important result and achievement of those presented here, from both, the academic and practical points of view; it is really not obvious that the duration of ascending and descending runs independently follow geometric distributions with for example different parameter values respectively.

From an academic point of view, it might be interesting to see what happens to the efficiency of markets when $p$ and $q$ differ significantly: would they adapt their price variations to

compensate for the difference in probabilities of seeing upward or downward uninterrupted trends? Answering this question would be material for another article. On the other hand, obviously these results could easily be incorporated into various trading systems. In addition, another simple application of our methodology, was ranking the different analyzed markets according to the larger fraction of time they follow the geometric behavior for the parametric case. Rank that, on the other hand, seems to coincide with the level of efficiency of the markets studied, issue that, since EMH is given in terms of market prices variations and not in terms of runs, also deserves further study.

Although discussed at last paragraph of section Introduction, we conclude this paper remarking that empirical results as those reported herein are also important and of interest because any adequate agents based market model or of any other kind must reproduce them. See references [4, 9].

## Supporting information

**S1 Dataset. File S1_DataSet.zip contains all analyzed data set.**
(ZIP)

**S1 Appendix. The discrete version of the Anderson-Darling goodness-of-fit test.**
(PDF)

## Acknowledgments

We thank Ms. S. Jiménez for her LATEX writing and correcting.

## Author Contributions

**Conceptualization:** Héctor Francisco Coronel-Brizio, Enrico Scalas, Thomas Henry Seligman, Alejandro Raúl Hernández-Montoya.

**Data curation:** Carlos Manuel Rodríguez-Martínez, Héctor Francisco Coronel-Brizio.

**Formal analysis:** Héctor Francisco Coronel-Brizio, Enrico Scalas, Thomas Henry Seligman.

**Funding acquisition:** Thomas Henry Seligman, Alejandro Raúl Hernández-Montoya.

**Investigation:** Héctor Raúl Olivares-Sánchez, Carlos Manuel Rodríguez-Martínez, Enrico Scalas, Thomas Henry Seligman, Alejandro Raúl Hernández-Montoya.

**Methodology:** Enrico Scalas, Thomas Henry Seligman, Alejandro Raúl Hernández-Montoya.

**Project administration:** Alejandro Raúl Hernández-Montoya.

**Software:** Héctor Raúl Olivares-Sánchez, Carlos Manuel Rodríguez-Martínez, Alejandro Raúl Hernández-Montoya.

**Supervision:** Héctor Francisco Coronel-Brizio, Thomas Henry Seligman, Alejandro Raúl Hernández-Montoya.

**Validation:** Enrico Scalas, Thomas Henry Seligman, Alejandro Raúl Hernández-Montoya.

**Visualization:** Carlos Manuel Rodríguez-Martínez.

**Writing – original draft:** Héctor Francisco Coronel-Brizio, Enrico Scalas, Thomas Henry Seligman, Alejandro Raúl Hernández-Montoya.

**Writing – review & editing:** Enrico Scalas, Alejandro Raúl Hernández-Montoya.

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
