## [Decision Letter · Decision Letter 0]

2 Dec 2021

PONE-D-21-27632An empirical data analysis of “price runs” in daily financial indices: dynamically assessing market geometric behaviorPLOS ONE

Dear Dr. Hernandez Montoya,

Thank you for submitting your manuscript to PLOS ONE. After careful consideration, we feel that it has merit but does not fully meet PLOS ONE’s publication criteria as it currently stands. Therefore, we invite you to submit a revised version of the manuscript that addresses the points raised during the review process.

We look forward to receiving your revised manuscript.

Kind regards,

Aurelio F. Bariviera, Ph.D.

Academic Editor

PLOS ONE

Journal Requirements:

"We thank Ms. Selene Jim´enez for her LATEX writing and correcting. This work has been endorsed by Conacyt-Mexico under project grant numbers 425854 and 5150 supported by FOINS. ES is partially supported by the Dr Perry James (Jim) Browne Research Centre at the Department of Mathematics, University of Sussex."

"ARHM and CMRM received support from grants 425854 and 5150 from the Consejo Nacional de Ciencia y Tecnología. Conacyt. https://conacyt.mx/

THS received support from grant number 425854 from the Consejo Nacional de Ciencia y Tecnología. Conacyt. https://conacyt.mx/

ES is partially supported by the Dr Perry James (Jim) Browne Research Centre at the Department of Mathematics, University of Sussex. http://www.sussex.ac.uk/broadcast/read/55282

Reviewers' comments:

Reviewer's Responses to Questions

**Comments to the Author**

1. Is the manuscript technically sound, and do the data support the conclusions?

Reviewer #1: Yes

Reviewer #2: Partly

2. Has the statistical analysis been performed appropriately and rigorously? 

Reviewer #1: Yes

Reviewer #2: I Don't Know

3. Have the authors made all data underlying the findings in their manuscript fully available?

Reviewer #1: Yes

Reviewer #2: No

4. Is the manuscript presented in an intelligible fashion and written in standard English?

Reviewer #1: Yes

Reviewer #2: Yes

5. Review Comments to the Author

Reviewer #1: The paper investigates the distribution of the duration of uninterrupted trends for the daily indices DJIA, NASDAQ, IPC, and Nikkei 225 from 10/30/1978 to 08/07/2020 and compares the simple geometric statistical model with p=1/2 consistent with the EMH to the empirical data. Results show that the geometric distribution with parameter p = 1/2 provides a good model for uninterrupted trends of short and medium duration for the more mature markets, however, the longest duration events still need to be statistically characterized.

As a general comment, I think the paper makes an interesting contribution to the literature by analyzing a new data analysis. At the same time, I think the paper needs some improvements before being published in this journal.

Specific comments

1) I suggest the authors number the sections

2) In section “Data sample and methodology” I would suggest the authors enrich the description by adding a Table reporting the main statistical properties of the data.

3) In the section conclusion, the authors mention that the empirical results found are also important for agent-based models to validate the simulations. I would suggest the authors add in the introduction a paragraph where the importance of stylized facts for markets simulators is described. I suggest also add the following citations:

a. Ponta, L., Trinh, M., Raberto, M., Scalas, E., & Cincotti, S. (2019). Modeling non-stationarities in high-frequency financial time series. Physica A: statistical mechanics and its applications, 521, 173-196.

b. Meyer, M. (2019). How to use and derive stylized facts for validating simulation models. In Computer simulation validation (pp. 383-403). Springer, Cham.

4) In the section conclusion, the authors should add the practical implication of the study.

5) Please check the quality of the figures. Some are very large others very small.

Reviewer #2: The authors do not show how they compute standard deviation of p and q, technically. The reviewer thinks that how to compute standard deviation of p and q is not obvious.

The title of this paper is not suitable to the current content proven in this manuscript. Specifically, the authors analyze the ratio of the direction of price changes by using p and q. Statistics of p and q are price direction for the short term (one business day), namely, Pr(x) with x = 1 or -1. The authors assume that the price direction should be independent. However, the price run should be measured by correlation among the directions of price change for some time horizon. Pr(x1, x2), Pr(x1, x2, x3), and more.

Such correlations are captured by test for independence by the chi-squared test for high dimensional contingency table.

Moreover, some researchers have been conducted this type of analysis in the literature of econophysics about fifteen years before. Please read the following book: Hideki Takayasu (ed.), Empirical science of financial fluctuations: the advent of econophysics, Springer (2002)

6. PLOS authors have the option to publish the peer review history of their article (what does this mean?). If published, this will include your full peer review and any attached files.

Reviewer #1: No

---

## [Author Response · Author response to Decision Letter 0]

11 May 2022

>Reviewer #1: The paper investigates the distribution of the duration of uninterrupted trends for the daily indices DJIA, NASDAQ, IPC, and Nikkei 225 from 10/30/1978 to 08/07/2020 and compares the >simple geometric statistical model with p=1/2 consistent with the EMH to the empirical data. Results show that the geometric distribution with parameter p = 1/2 provides a good model for uninterrupted >trends of short and medium duration for the more mature markets, however, the longest duration events still need to be statistically characterized.

>As a general comment, I think the paper makes an interesting contribution to the literature by analyzing a new data analysis. At the same time, I think the paper needs some improvements before being >published in this journal.

Thank you very much for your opinion on our work.

>Specific comments

>1) I suggest the authors number the sections

To the best of our knowledge, Plos ONE journal does not include enumeration of sections, subsections, etc. in its published articles. This is easy to check by downloading any issue from Plos ONE journal website. We are only following Plos ONE latex template to write and submit our paper. 

>2) In section “Data sample and methodology” I would suggest the authors enrich the description by adding a Table reporting the main statistical properties of the data.

We agree with this suggestion, we have included in our paper the new table number 6, displaying the descriptive statistics of the analyzed data and edited the text accordingly.

>3) In the section conclusion, the authors mention that the empirical results found are also important for agent-based models to validate the simulations. I would suggest the authors add in the >introduction a paragraph where the importance of stylized facts for markets simulators is described. I suggest also add the following citations:

We agree with this suggestion, in section “Introduction”, we have added the following text (lines 27 to 37): 

“Empirical studies of financial and economic data are becoming increasingly relevant for the following reasons: 

1) Currently dozens of stylized facts have been observed and more are still being discovered.

2) The study and prediction of stylized facts by means of methodologies of multi-agents market models is an important area of research in Finance and Econophysics.

3) Stylized facts are an import tool to validate proposed numerical and multi-agent market models; and

4) At present, we still lack a general, microscopic theory or model to explain the origin of stylized facts, we think simulation methodologies using agents could be useful in the construction of such general theory. Some interesting references on these issues are the following: [3–8].”

Where the corresponding cited bibliography is:

3. Pagan, A., The econometrics of financial markets. Journal of Empirical Finance. 1996;3(1):15–102. doi: 1016/0927-5398(95)00020-8.

4. Ponta L, Trinh M, Raberto M, Scalas E, Cincotti S. Modeling non-stationarities in high-frequency financial time series. Physica A, 2019;521(1):173–196. doi: 10.1016/j.physa.2019.01.069.

5. Maldarella D, Pareschi L. Kinetic models for socio-economic dynamics of speculative markets. Physica A, 2012;391(1):715–730. doi: 10.1016/j.physa.2011.08.013.

6. Ehrentreich N. Agent-based modeling: The Santa Fe Institute artificial stock market model revisited. Volume 602. ISBN: 978-3-540-73879-4, Springer Science & Business Media (2007).

7. Lux T. Stochastic behavioral asset pricing models and the stylized facts. Technical report. 2008, Economics working paper, Department of Economics. Christian Albrechts-Universitat Kiel.

8. Farmer JD. The economy needs agent-based modeling. Nature. 2009;460(7256):685–686. doi: 10.1038/460685a.

9. Meyer M. How to Use and Derive Stylized Facts for Validating Simulation Models. In Claus Beisbart and Nicole J. Saam (eds.), Computer Simulation Validation: Fundamental Concepts, Methodological 

Frameworks, and Philosophical Perspectives. Series: Simulation Foundations, Methods and Applications. ISBN: 978-3319707655, Springer. pp. 383-403 (2019).

10. Takayasu H. (Ed), Empirical Science of Financial Fluctuations: The advent of econophysics, ISBN: 978-4431703167, Springer (2000).

>4) In the section conclusion, the authors should add the practical implication of the study.

We thought we had explained the main applications of our results in the "Results" section; however, this question gives us the opportunity to think about this issue and make explicit what we consider to be the most important implication of our results. So, as requested, we have added the following text in the "Conclusions" section, lines 497 to 518:

“we show their time evolution as well as the time evolution of their addition $p+q$. Moreover, we not only show that the runs distribution of the markets studied is compatible with the geometric distribution with the estimated parameters, but we also estimate when and the fraction of the time during which the markets follow this behavior, parametrically for $p=q=0.5$ and non-parametrically. The detection of when and for how long the distribution of the durations of market price runs have a geometric distribution is in our opinion our most important result and achievement of those presented here, from both, the academic and practical points of view; it is really not obvious that the duration of ascending and descending runs independently follow geometric distributions with for example different parameter values respectively. 

From an academic point of view, it might be interesting to see what happens to the efficiency of markets when $p$ and $q$ differ significantly: would they adapt their price variations to compensate for the difference in probabilities of seeing upward or downward uninterrupted trends? Answering this question would be material for another article. On the other hand, obviously these results could easily be incorporated into various trading systems. In addition, another simple application of our methodology, was to classify the different analyzed markets according to the largest fraction of time they follow the geometric behavior for the parametric case. Rank that, on the other hand, seems to coincide with the level of efficiency of the markets studied, issue that, since EMH is given in terms of market prices variations and not in terms of runs, also deserves further study.”

>5) Please check the quality of the figures. Some are very large others very small.

I am afraid, just as the same case of point 1), we do not have control on this issue, Plos ONE latex template for papers to be evaluated by reviewers gives this output. By using the Plos ONE final version latex template, all figures look good.

>Reviewer #2: The authors do not show how they compute standard deviation of p and q, technically. The reviewer thinks that how to compute standard deviation of p and q is not obvious.

We agree with your comment, we have added in pages 19-20, lines 328 to 356, a new subsection titled “Estimation of $<\\hat{p}>$ and $\\sigma_p$”, where we explain the methodology for the calculation of parameter $p$. We have added the following references to the text:

37. Silvey SD. Statistical Inference. Chapman & Hall. London. Chapman & Hall Monographs on Statistics and Applied Probability; 1975. ISBN: 978-0412138201. London.

38. Bickel PJ, Doksum KA. Mathematical Statistics ISBN: 978-0816207848. Prentice Hall, Englewood Cliffs, New Jersey; 1977.

39. Harding B, Tremblay C, Cousineau D. Standard errors: A review and evaluation of standard error estimators using Monte Carlo simulations TQMP. 2007; 10(2):107–123. 

doi:10.20982/tqmp.10.2.p107.

>The title of this paper is not suitable to the current content proven in this manuscript.

We agree in this point, the title could be misleading. For this reason, we have included in it the word “distributional” to have the following final title: 

“An empirical data analysis of “price runs” in daily financial indices: dynamically assessing market geometric distributional behavior”.

It is a subtle change, but in this, we gain clarity about what we are really analyzing.

>Specifically, the authors analyze the ratio of the direction of price changes by using p and q. Statistics of p and q are price direction for the short term (one business day), namely, Pr(x) with x = 1 or -1. >The authors assume that the price direction should be independent. 

In reality, we do not analyze explicitly the ratio of the direction of price changes, in fact, in order to motivate presented ideas, we only show a single plot showing the behavior of ratio of upward to total prices evolution. In our paper we count and directly analyze upward and downward run durations separately, assuming they are independent, instead of assuming the condition $p+q = 1$; on the contrary we statistically verify the condition $p+q = 1$.

>However, the price run should be measured by correlation among the directions of price change for some time horizon. Pr(x1, x2), Pr(x1, x2, x3), and more. Such correlations are captured by test for >independence by the chi-squared test for high dimensional contingency table.

We are not sure about the meaning of “price runs measurement”, in our paper we study the distribution of price runs length or price runs duration. In any case, if the suggestion is to study the finite dimensional distributions of up and down price movements, this would be a different analysis, as we are focusing on runs.

By the other side, we have studied autocorrelations of returns coming from runs. This is more natural because returns calculated from runs are signed while runs durations are not. However, this would be also a different analysis than the presented in our paper.

We consider that as it stands our paper is already sufficiently large and complete to include additional analyses. However, the idea of using contingency tables to analyze data seems really interesting to us and surely is worth of exploration, but again this would be a different analysis than the one presented in our article.

>Moreover, some researchers have been conducted this type of analysis in the literature of econophysics about fifteen years before. Please read the following book: Hideki Takayasu (ed.), Empirical >science of financial fluctuations: the advent of econophysics, Springer (2002)

In fact, these kind of studies and other related, as for example returns distribution or in general, the study of market empirical properties, have a long tradition in the econometric literature; are currently actively investigated for a great number of researchers worldwide, and date back much more than 15 years ago (our older reference dates back to 1937), as we show in our bibliography. 

Universal empirical properties of financial data, named stylized facts, is an active research area for the economics and physicists communities, and dozens of these properties have been reported as well as some new others.

Returning to our work, we should stress that we report empirical results, which to our knowledge are novel and original. For example, as we answered to referee number one in above point 4), no methodology and measurement of the fraction of time in which the distribution of market price runs follows a geometric distribution with parameter compatible with p=0.5 or the dates when this happens has been published before, the same can be said of the result that geometric distribution is a good model of the duration of the runs with the parameter p changing or of how p and q evolve with time, etc. In opinion, these are enough interesting for the academical community.

Thank you for the reference, we have included the outstanding book edited by Professor Takayasu in our bibliography. It includes papers describing important methodologies that would become of great importance to study financial markets, such as the applications of agents and to model financial markets or the applications of random matrix theory to study the correlations of market sectors, etc.

It has been included as reference number 10, lines 557 and 558:

10. Takayasu H. (Ed), Empirical Science of Financial Fluctuations: The advent of econophysics, ISBN: 978-4431703167, Springer (2000).

However, we finally mention here, that this reference does not include any work with an analysis similar to the one presented by us in our paper. 

On code and data availability:

Mathematica and python versions of the code used in our data analyses presented in our paper, may be downloaded at:

https://github.com/CarlosManuelRodr/TrendDurationAnalysis

Analyzed data is available at:

https://github.com/CarlosManuelRodr/TrendDurationAnalysis/tree/main/Research/OriginalDataset

This is now mentioned in the text, lines 167 to 169:

“Data sample files were downloaded from Yahoo Finance. All data sample for the mentioned time span is available at the following link: https://github.com/CarlosManuelRodr/TrendDurationAnalysis/tree/main/Research/OriginalDataset.”

Finally, a regular number of minor corrections were made to the manuscript, and they are:

i) Table 3, pg 10: For Nasdaq uptrends with durations 16 days, entry must be 0 instead 1 and for Nasdaq downtrends with duration 16 days entry must be 1 and not 0. For downtrends with durations 18 and 19 days, both entries must be 0 and not 1. 

From figures 4c) and 4d), it is clear that above errors were really typos. It can be seen there that distribution of Nasdaq uptrends durations has no entries with 16 days, and there is an entry in Nasdaq downtrends durations distribution with duration of 16 days and no entries for downtrends with durations 18 and 19 days because in this case the maximum downtrend duration is 16 days long.

For convenience to referees to verify these typos in table 3, below we have inserted subfigures 4c) and 4d) of our paper: 

(Here we insert subfigures 4c) and 4d) in the pdf file of "Response to Reviewers letter".)

 ii) These simple corrections imply that we should have corrected in same table 3 the total number of uptrends in second column, first row from 2390 to 2389; the total number of downtrends in first row third column from 2391 to 2390 and the overall number of trends in fourth column from 4781 to 4779.

iii) For same reason, and for Nasdaq, table 1 third row, columns 3rd, 4th and 5th were corrected from 4781, 2390 and 2391 to 4779, 2389 and 2390 trends, respectively. Also for Nasdaq, number of records has been corrected from 10481 to 10534 entries. The correctness of the last change may be verified observing the number of records in file:

https://github.com/CarlosManuelRodr/TrendDurationAnalysis/blob/main/Research/OriginalDataset/nasdaq.json

which has 10549 records in total, with 13 initial lines with information on the index itself, time period covered and two final lines with no data, remaining a total of 10534 Nasdaq data records to analize.

Finally, to indicate some data preprocessing was applied, the text: “The data have been filtered, e.g. by removing null records.” has been added to the captions of figure 1 (for example, Nikkei data has eleven null records). 

iv) In tables 2 to 5, first rows, first columns: the units of the variable duration have been included  (days).

v) In line 24:

It said;

“...where we describe a basic random process consistent…”

It says now:

“...where we study empirically a basic random process consistent …”

vi) Because its historical and theoretical interest, he following text was inserted in lines 151 to 157:

“Some historical references on this subject, are [19], [20] and [21]. Where chapter X of the first reference was during many years the classical textbook reference to Theory of Runs; second reference shows an interesting statistical test based on runs properties to demonstrate that two sets of independent observations corresponding to two independent random variables have the same distribution and finally, the third reference presents an intensive treatment of the theory of runs still of current interest.”

Respective, included references are:

19. Wilks SS. Mathematical Statistics, ISBN: 978-4431703167, Princeton University Press (1943).

Available from: https://books.google.com.mx/books?id=k38pAQAAMAAJ}. 

20. Wald A, Wolfowitz J. On a test whether two samples are from the same population. Ann. Math. Stat. 1940;11:147–162. Available from: http://dml.mathdoc.fr/item/1177731909/.

21. Mood AM. The Distribution Theory of Runs. Ann. Math. Stat. 1940;11:367–392. Available from: https://projecteuclid.org/journals/annals-of-mathematical-statistics/volume-11/issue-4/The-Distribution-Theory-of-Runs/10.1214/aoms/1177731825.full

vii) In line 487:

“…financial crahes…” was corrected to “…financial crashes...”

viii) In reference 1. line 532, “Lo AWC...” was corrected to “Lo AW...”

ix) Reference 22 in lines 583 to 585 is not available online any more, then inactive URL has been deleted, also in S1.Appendix Supporting information.

x) In reference 32, lines 605 and 606, Kertez was corrected to Kertész. Also, year of reference was corrected from 2005 to 2006.

xi) In references 33) and 34), in lines 607 and 610, respectively, “et all” was corrected to “et al”.

Again, we thank you and our anonymous referees for your time and effort in reviewing our paper.

---

## [Decision Letter · Decision Letter 1]

13 Jun 2022

An empirical data analysis of “price runs” in daily financial indices: dynamically assessing market geometric distributional behavior

PONE-D-21-27632R1

Dear Dr. Hernandez Montoya,

We’re pleased to inform you that your manuscript has been judged scientifically suitable for publication and will be formally accepted for publication once it meets all outstanding technical requirements.

Kind regards,

Aurelio F. Bariviera, Ph.D.

Academic Editor

PLOS ONE

Additional Editor Comments (optional):

Reviewers' comments:

Reviewer's Responses to Questions

**Comments to the Author**

1. If the authors have adequately addressed your comments raised in a previous round of review and you feel that this manuscript is now acceptable for publication, you may indicate that here to bypass the “Comments to the Author” section, enter your conflict of interest statement in the “Confidential to Editor” section, and submit your "Accept" recommendation.

Reviewer #1: All comments have been addressed

2. Is the manuscript technically sound, and do the data support the conclusions?

Reviewer #1: Yes

3. Has the statistical analysis been performed appropriately and rigorously? 

Reviewer #1: Yes

4. Have the authors made all data underlying the findings in their manuscript fully available?

Reviewer #1: Yes

5. Is the manuscript presented in an intelligible fashion and written in standard English?

Reviewer #1: Yes

6. Review Comments to the Author

Reviewer #1: The paper has been improved following the referees’ suggestions, and now, according to me, it is ready to be published in this journal.

7. PLOS authors have the option to publish the peer review history of their article (what does this mean?). If published, this will include your full peer review and any attached files.

Reviewer #1: No

---

## [Editor Report · Acceptance letter]

23 Jun 2022

PONE-D-21-27632R1 

An empirical data analysis of “price runs” in daily financial indices: dynamically assessing market geometric distributional behavior 

Dear Dr. Hernández-Montoya:

I'm pleased to inform you that your manuscript has been deemed suitable for publication in PLOS ONE. Congratulations! Your manuscript is now with our production department. 

Kind regards, 

on behalf of

Dr. Aurelio F. Bariviera 

Academic Editor

PLOS ONE